# Conductive Polymers and Their Nanocomposites: Application Features in Biosensors and Biofuel Cells

**DOI:** 10.3390/polym15183783

**Published:** 2023-09-15

**Authors:** Lyubov S. Kuznetsova, Vyacheslav A. Arlyapov, Yulia V. Plekhanova, Sergei E. Tarasov, Anna S. Kharkova, Evgeniya A. Saverina, Anatoly N. Reshetilov

**Affiliations:** 1Federal State Budgetary Educational Institution of Higher Education, Tula State University, 300012 Tula, Russia; 2Federal Research Center «Pushchino Scientific Center for Biological Research of the Russian Academy of Sciences», G.K. Skryabin Institute of Biochemistry and Physiology of Microorganisms, Russian Academy of Sciences, 142290 Pushchino, Russia; 3Federal State Budgetary Institution of Science, N.D. Zelinsky Institute of Organic Chemistry, 119991 Moscow, Russia

**Keywords:** conducting polymers, nanocomposites, biosensors, microbial and enzymatic biofuel cells, polymer-modified electrodes, electrochemical sensors, bioelectrochemistry

## Abstract

Conductive polymers and their composites are excellent materials for coupling biological materials and electrodes in bioelectrochemical systems. It is assumed that their relevance and introduction to the field of bioelectrochemical devices will only grow due to their tunable conductivity, easy modification, and biocompatibility. This review analyzes the main trends and trends in the development of the methodology for the application of conductive polymers and their use in biosensors and biofuel elements, as well as describes their future prospects. Approaches to the synthesis of such materials and the peculiarities of obtaining their nanocomposites are presented. Special emphasis is placed on the features of the interfaces of such materials with biological objects.

## 1. Introduction

Conductive polymers (CPs) are a class of organic compounds that belong to a large group of electrochemically active polymers [1]. They have been actively studied for more than 40 years, beginning with the discovery of the conductive properties of halogen-substituted polyacetylene [2] and the preparation of polypyrrole [3]. CPs are an important class of functional materials that have found wide application in many areas of science [4]. They have excellent properties, such as adjustable conductivity, ease of application to the electrode surface, and the possibility of simple chemical modification. Due to their biocompatibility, they are used in the field of regenerative medicine [5,6], tissue engineering [7,8], electrically induced targeted drug release systems [9,10], and cell cultivation [11]. It is also possible to use conductive materials in energy storage devices, electronic devices, and anticorrosion coatings [12,13,14,15,16,17]. Due to their high conductivity, such polymers and composites based on them are excellent materials for the conjugation of biological materials and electrodes in bioelectrochemical systems [18,19,20]. At the same time, conductive polymers can act as effective matrices for the immobilization of biological materials, enabling the free diffusion of substrates and reaction products and providing a biocompatible environment for enzymes or microorganism cells (Figure 1).

Electrically conductive polymeric materials are divided into two large groups. The first group includes polymers with ionic conductivity, or solid polymer electrolytes, and the second group includes polymers with electronic conductivity. In turn, polymers with electronic conductivity are divided into so-called organic metals (polymers with a conductivity similar in mechanism to the electrical conductivity of metals and called “conducting polymers”) and redox polymers (compounds in which electron transfer is carried out mainly due to the flow of oxidative–reduction reactions between neighboring fragments of the polymer chain) [21].

The electronic conductivity of conducting polymers exists due to delocalized electrons (n-conductivity) or holes (p-conductivity) (a unit charge in such polymers, as a rule, is delocalized over several fragments of the polymer chain). The conductivity of conducting polymers is due to the generation of charge carriers (polarons and bipolarons) in the polymer phase as a result of the oxidation of double bonds and heteroatoms with unshared electron pairs [4]. Polarons and bipolarons are radical cations and dications delocalized in the polymer chain. The delocalization of the charge and unpaired electrons in polarons and bipolarons leads to the structural deformation of the polymer lattice, while the stabilization of polaron charge carriers occurs due to polarization and interaction with counterions. During the migration of polarons and bipolarons along the polymer chain, the electronic conductivity of polymers is realized by the reorganization of single and double bonds. In defective areas, charge transfer is carried out due to interchain activation “jumps” of electrons, that is, the “hopping mechanism”.

Redox-active polymers are also a class of electrochemically active compounds; however, they differ due to the presence of discrete redox-active centers [22]. The process of electron transfer in the polymer layer can be represented as a sequence of reactions involving the self-exchange of electrons between redox centers in different oxidation states. The conductivity of the polymer reaches its maximum value when the volume concentrations of oxidized and reduced fragments are equal, namely, at the formal potential of the redox system. Both redox-active and conductive ones have a mixed electronic–ionic conduction mechanism, which includes the interfacial transfer of electrons and ions through the phase boundaries and the conjugate electron–ion transfer in the bulk of the material. The charge transfer in the electrically conductive layer is accompanied by the diffusion of supporting electrolyte ions into the bulk of the polymer to maintain its electrical neutrality. To describe the charge transfer in redox-active and conducting polymers, the same model of the “hopping mechanism” accompanied by ion transfer is used [4]. Polymerized phenazines (poly(thionine), poly(neutral red), poly(methylene blue), etc.) occupy a special place in the group of electron-conducting polymers. These phenazines have delocalized π electrons in their structure, like conducting polymers; however, during polymerization, they retain electron transfer in separate monomeric fragments, and, therefore, they are commonly referred to as redox-active polymers [23,24,25,26].

The electrical properties of conducting polymers can be varied by doping, increasing the degree of charge carriers [27], which, in turn, will increase their applications, particularly in bioanalytical and bioelectrochemical devices. In the undoped state, conducting polymers behave as anisotropic quasi-one-dimensional electronic structures with a moderate bandgap of 2–3 eV, as in a conventional semiconductor. When a conductive polymer is doped or photoexcited, the polymer changes to a “metallic” state. In recent decades, nanomaterials have been the most commonly used dopants due to their excellent electrocatalytic properties, multiple active sites, structural properties, and good stability. Such nanomaterials can also be effectively used in combination with conductive polymers in the modification of biosensor electrodes or BFCs. Simplification of the conditions for the synthesis of various nanomaterials has led to an increase in the number of works in the field of nanocomposites with conductive polymers in the last 10 years (Figure 2).

This review analyzes the main trends in and prospects for the development of the methodology for the use of conductive polymers in such bioelectrochemical devices as biosensors and biofuel cells. Approaches to the synthesis of such materials and the specifics of obtaining their nanocomposites are presented. Particular attention is paid to the features of the conjugation of such conductive materials with biological objects.

## 2. Synthesis of Conducting Polymers

The most widely used conductive polymers are polyacetylene, polyaniline, polypyrrole, polythiophene, poly(para-phenylene), poly(phenylvinylene), and polyfuran (Figure 3). Their electrical properties are presented in Table 1.

One of the most successful polymers in terms of synthesis is PEDOT:PSS (the synthesis is patented by Bayer). Its production reaches more than 100 tons per year. The production of polypyrrole and polyaniline is also widely scaled [31]. Conductive polymers are obtained in several ways, the main one of which is polymerization. Polymerization consists of three main steps: oxidation, binding, and deprotonation (Figure 4). 

The process begins with the oxidation of monomer units to form radical cations, which then combine with each other or with other monomers. Bound cations are deprotonated, and dimer fragments are formed [32,33]. The process of oxidation and binding is repeated with dimers, leading to an increase in the chain and the formation of long chains of polymers [15]. There are several ways to initiate polymerization, such as chemical, electrochemical, and photopolymerization.

### 2.1. Chemical Synthesis of Conductive Polymers

Chemical polymerization is one of the most widely used methods for producing conductive polymers, along with electropolymerization. The method is scalable and economical and does not require expensive equipment. Usually, in chemical synthesis, the polymer has fewer internal crosslinks between macromolecules, and therefore, it is more soluble in suitable solvents. In addition, this method has more opportunities for the covalent modification of conducting polymers [34]. However, chemical polymerization does not always ensure the high electrical conductivity of the obtained polymers, does not allow for controlling the thickness of coatings, and is more suitable for obtaining thick-film and powder polymers. Also, oxidizing agents are required for the synthesis by this method, which can lead to contamination of the final product [35].

Chemical polymerization is carried out using various oxidizing agents, such as ammonium persulfate ((NH_4_)_2_S_2_O_8_), sodium vanadate (NaVO_3_), cerium sulfate (Ce(SO_4_)_2_), hydrogen peroxide (H_2_O_2_), potassium iodate (KIO_3_), potassium dichromate (K_2_Cr_2_O_7_), and hydrochloric acid (HCl_4_) [36,37,38,39]. The mechanism of chemical polymerization, using polypyrrole as an example, is shown in Figure 5.

The use of the catalytic action of redox enzymes makes it possible to carry out chemical polymerization under milder and environmentally friendly conditions. For example, peroxidase and laccase are used to synthesize polyaniline and polypyrrole [41]. The review in [42] considers the latest achievements over the last 10 years in the synthesis of polyaniline using laccase with atmospheric oxygen, and it is shown that this synthesis method is variable in terms of settings that can be controlled using templates. Figure 6 shows how the laccase-mediated synthesis of polyaniline from aniline monomers and p-aminodiphenylamine occurs.

Interfacial polymerization is an example of chemical oxidative polymerization that has found wide application in the preparation of conductive polymers for chemical sensors, fuel cells, supercapacitors, and membranes with selective permeability. This method includes several types of polymerization with phase separations: liquid–solid, liquid–liquid, and liquid-in-liquid (Figure 7).

Reference [44] reports that PEDOT obtained by interfacial polymerization acquires unique properties, such as a high porosity of 70.61%, a high specific surface area > 58 m^2^/g, and an ideal electrical conductivity of 6500 S/m, which provides a wide voltage range for supercapacitors (up to 1.2 V). The material demonstrates outstanding energy storage characteristics in electronics, while the proposed method is environmentally friendly (Figure 8).

The method is widely used to obtain polyaniline (PANI) [45,46], polypyrrole (PPy) [45,47], poly(3,4-ethylenedioxythiophene) [44], and polythiophene (PTh) [48,49] and nanofibers based on them, as well as nanocomposites. 

### 2.2. Electropolymerization of Conductive Polymers

Electrochemical polymerization is the most commonly used method for the synthesis of conductive polymers because it allows better control of polymer deposition, film morphology, electrical conductivity, and mechanical properties. In addition, this method allows the direct application of the polymer to the substrate/substrate. This method is not suitable for all types of conductive polymers. In addition, it is difficult to scale and can lead to the irreversible oxidation of monomers due to the application of high potentials [34]. Electrochemical polymerization is carried out using an electrochemical cell with a three-electrode system containing working, counter, and reference electrodes. The solution typically contains monomers, a solvent, and an electrolyte/salt. Polymerization can be initiated using cyclic voltammetry, potentiostatically, or using a galvanostatic method. The potentiostatic method makes it easy to control the film thickness using Faraday’s law, while potentiodynamic methods lead to the formation of more uniform and adhesive films on the electrode. The galvanostatic method is generally considered more efficient because it allows the growth of the conductive polymer film to be monitored and the change in conductivity during the polymerization process.

The method is widely used for the synthesis of many known conductive polymers, for example, polyaniline (Figure 9) [50], polypyrrole [51,52], and PEDOT [37].

### 2.3. Photochemical Polymerization of Conductive Polymers

Photopolymerization is a special form of radical polymerization that uses light to initiate polymerization. Gamma rays [53], microwaves [54], ultraviolet rays [55], and X-rays [55] are used as sources of radiation. Moreover, the synthesis of conductive polymers proceeds indirectly using photosensitizers. In this case, the transfer of light energy occurs through a photosensitizer to form the corresponding excited states. Ruthenium complexes, silver nitrate, camphorquinone, and ketones are used as photosensitizers. For photoelectrochemical polymerization, the photosensitizer is a dye-sensitized semiconductor (metal oxides such as TiO_2_, ZnO, and WO_3_; chalcogenides such as CdS, CdSe, and GaAs) or simply a dye.

Photochemical polymerization takes place at lower temperatures than chemical polymerization and allows the control of the reaction by simply turning on/off the light source [55,56]. This method solves the problem of excessive oxidation of the obtained polymers and does not harm the environment; however, not all polymers can be obtained by this method [28].

The main limitation of light-induced polymerization is the insignificant depth of light penetration, which does not exceed a few millimeters and depends on the wavelength and spectral distribution. Photopolymerization techniques are only applicable to thin-film applications because polymerization stops when the light source is removed. This method is used for the synthesis of polypyrrole [57], polyaniline, and hybrid materials based on polyaniline [58], as well as some other conductive polymers.

### 2.4. Synthesis of Conductive Nanocomposites

The use of pure polymers is limited; therefore, their composites with various compounds are obtained. For example, the synthesis of nanocomposites can occur in situ (sequentially), ex situ (separately), and in one pot (simultaneously). Approaches to the synthesis of nanocomposites can be divided into synthesis using templates, template-free synthesis, and other methods (Figure 10). They are described in detail in [28,34,59].

The template method involves applying hard or soft templates. Solid templates are used as membranes to obtain nanostructures inside the pores or channels of the membrane to control the morphology of the future polymer. Carbon nanotubes, graphite, aluminum oxide, and vanadium are described as templates. The synthesis of nanopolymers is carried out chemically and electrochemically. The disadvantages of this approach include the fact that it is limited in terms of production scale, templates are needed for synthesis, and the use of aggressive reagents is required to remove templates after the reaction. An example of obtaining nanotubular polyaniline using natural halloysite as a template is shown in Figure 11.

The solution to these problems can be a soft template method based on the use of surfactants, micelles, and structure-forming molecules, which are usually created using self-assembly mechanisms using hydrogen bonds, etc. The morphology of the obtained conducting polymers directly depends on the morphology and molecular matrices of the templates. The method is distinguished by its simplicity, its low cost, and the ability to carry out synthesis on a large scale. However, the disadvantages include the difficulty of controlling the size and orientation of the conductive structures. This method is used for the synthesis of polyaniline [61]. The article considers the influence of anionic polyelectrolytes, micelles, and vesicles as matrices for obtaining conductive polymers. It has been shown that obtaining polyaniline with a higher content of the polaron form is observed when using vesicles from sodium bis(2-ethylhexyl)sulfosuccinate as a matrix (Figure 12).

The templateless method allows one to bypass the limitations of the templated method. In this case, combined materials are used to obtain a highly organized architecture of finished products. Synthesis is controlled by varying the temperature and the ratio of the monomer and dopant. Commonly used zero-template technologies include electrospinning, seeding, interfacial polymerization (which is described above), and other nanofabrication methods [54].

Electrospinning is a universal method that allows for obtaining ultrathin fibers as a result of the action of electrostatic forces on an electrically charged jet of a polymer solution or melt. The electrospinning method makes it possible to obtain polymer fibers with a diameter of several hundred nanometers. The formation of ultrathin fibers by electrospinning is carried out by the uniaxial electrical stretching of a viscoelastic solution. Electrospinning requires three main components: a high-voltage power supply, a solution container with a spinneret, and a grounded metal collector [62].

The study in [63] describes a one-step method of polypyrrole polymerization and in situ immobilization in a poly(ε-caprolactone) polymer matrix to develop an efficient conductive biomaterial that has a potential result for bone tissue engineering. The increased surface-area-to-volume ratio due to the fine nanofiber morphology, the conductive properties of the highly conjugated PPy backbones, and improved mechanical strength, surface wettability, and biological activity contributed to better cell attachment and proliferation in PCL/PPy conductive scaffolds. 

Comparing all of the methods for the synthesis of conductive polymers and their nanocomposites discussed above, it is obvious that the possibilities for the synthesis of conductive polymers and nanocomposites are enormous. Strategies for integrating nanostructures into a polymer matrix with the desired strong interfacial interactions have further advanced this unique group of materials that will find wide application. At the same time, the possibilities for synthesis may develop toward the appearance of new monomers with higher conductivity. An increasingly important role in the synthesis of nanocomposites will be played by a controlled nanoarchitecture, which will allow the structures and properties of future nanocomposites to be set with high accuracy, as well as the inclusion of ionic liquids that improve the synthesis process and the quality of the materials obtained. The movement of synthesis toward greater environmental friendliness will be quite relevant: the use of biological templates and the use of methods with a lower load on the environment. These trends in the synthesis of conductive polymers and their nanocomposites make it possible to obtain more efficient and functional materials with a wide range of capabilities in various fields, including electronics, energy, medicine, sensorics, and others.

## 3. Composite Materials Based on Conductive Polymers

The combination of polymers with nanoparticles forms new electrical and catalytic properties, contributing to the improvement of biosensor devices and biofuel cells based on them [64,65].

### 3.1. Composites Based on Carbon Nanomaterials and Conductive Polymers

Among various nanocomposite materials, much attention is paid to composites based on carbon nanomaterials and conductive polymers. Carbon nanomaterials include all allotropic forms of carbon having a transverse size in the range of nanometers (1–100 nm). Due to their unique electrical, thermal, chemical, and mechanical properties, such composites have found application in areas related to storage, energy conversion, and biosensors. Moreover, the possibility of synthesis through functional groups or the formation of three-dimensional arrays allows the development of catalysts with a large surface area and materials with high electrochemical activity. The use of nanomaterials can help solve some of the key problems in the development of biosensors, such as more sensitive registration of the interaction of an analyte with the biosensor surface and a reduction in response time.

The following carbon nanomaterials are used to form hybrids used to develop biosensors:Fullerenes are spherical molecules in which carbon atoms are connected to each other through pyramidal hybrid sp^2^–sp^3^ orbitals.Graphene is a single layer of graphite that is an atom thick, where the carbon atoms have sp^2^ hybridization and are arranged in a honeycomb pattern. Graphene derivatives:
Graphene oxide is functionalized graphene with oxygen-containing functional groups;Reduced graphene oxide is treated graphene oxide with reduced oxygen content. The complete reduction of graphene oxide does not lead to the formation of a graphene layer due to residual oxygen-containing functional groups, since not all sp^3^ bonds return back to the sp^2^ configuration.
Single-walled carbon nanotubes (SWCNTs) or graphene nanotubes are graphene planes rolled into a cylinder, and multi-walled carbon nanotubes (MWCNTs) are a set of cylinders with different diameters nested into each other.Carbon nanodots or carbon quantum dots are quasi-spherical nanoparticles less than 10 nm in size containing various functional groups (carboxyl, amino groups).Graphene nanoribbons are strips of graphene less than 100 nm wide.

By analyzing the data on publications in the field of biosensor applications of composites based on conductive polymers and carbon nanomaterials (Figure 13), we can conclude that the most common are composites based on graphene and nanotubes, while their total number is steadily growing.

In [66], a new modified method of atom transfer radical polymerization was proposed to create a nanostructured material that is promising for energy storage (Figure 14).

The synthesis proposed by the authors makes it possible to control the polymerization rate and create new nanostructured materials consisting of CNT bundles wrapped in polypyrrole (Figure 15).

The authors of [67] proposed a simple process of self-assembly of a three-dimensional porous composite based on poly(3,4-ethylenedioxythiophene) (PEDOT), poly(4-styrenesulfonate) (PSS), and graphene (Figure 16). 

Graphene oxide (GO) sheets have many oxygen-containing groups (hydroxyl, carboxyl, and epoxy) with a negative charge. Due to electrostatic attraction, binding occurs between the positively charged PEDOT chains, the negatively charged PSS chains, the negatively charged GO sheets, and the positively charged hydronium ion. After the addition of HI, the electrostatic attraction between PEDOT and PSS is reduced, and H^+^ ions can bind to the negative chains of PSS to form hydrophilic PSSH chains. As a result, the PEDOT chains are stretched, resulting in an increase in conductance. At the same time, with the help of HI, graphene oxide is reduced, and the number of oxygen-containing functional groups in rGO graphene decreases. The resulting composite has excellent specific capacity, good mechanical strength, and extensibility and can be used in energy storage devices.

Carbon nanodots are another type of nanomaterial used to improve the electrochemical properties of polymers. In [68], a noncovalent hybrid of polypyrrole and carbon nanodots was obtained (Figure 17).

The synthesis of the pyrrole monomer in the presence of the surfactant decyltrimethylammonium bromide (DeTAB) was carried out at 0 °C. The addition of an ammonium persulfate (APS) oxidizing agent to the reaction mixture caused the start of the polymerization process. The last and most important step was the introduction of carbon nanodots or GQDs into the reaction mixture. The synthesized hybrids were modified polymer spheres of carbon nanodots coated with polypyrrole (Figure 18).

The resulting noncovalent system is based on the occurrence of interactions between carbon nanodot rings and heterocyclic systems present in the polypyrrole structure. The presence of functional groups at the edges of carbon nanoparticles increases their hydrophilicity. The mutual electrostatic repulsion of the structures allows them to act as an additional surfactant, which increases the colloidal stability of the reaction mixture.

Fullerenes C_60_ are widely used to design electrochemical biosensors. One of the promising methods for obtaining nanocomposites based on fullerene is polymerization, in which the resulting fullerene units are connected by polymer side chains or through the formation of epoxides [69,70,71,72]. The formation of a nanohybrid based on a polyamidoamine polymer with a terminal amino group and a highly branched structure for the modification of fullerene and molybdenum disulfide was shown in [70]. This modification (Figure 19) made it possible to improve the hydrophilic properties and increase the number of biomolecules on the electrode. A DNA biosensor based on the resulting nanohybrid was used to detect the specific *Sul1* gene of pathogenic bacteria *Salmonella typhimurium* with a linear range from 40 fM to 40 nM.

The authors of [71] describe ultrathin (80–100 nm) composite films based on poly(ethylene glycol) and fullerene; due to their excellent electrochemical properties, these materials can be used in biosensor devices. The composite films are made by crosslinking polyethylene glycol with an amino/epoxy resin. In this case, fullerene clusters with an average diameter of 314–329 nm can be located on the surfaces of the films, embedded in their volume, or physically or chemically combined with the polymer network.

A bilayer lipid membrane modified with fullerene was used to immobilize β-lactamase in [73]. The bilayer membrane (Figure 20) is based on cholesterol and lecithin dissolved in n-hexane. The membrane prevents hydrophilic electroactive substances from reaching the transducer, which reduces signal distortion. The biosensor allows the determination of benzylpenicillin in the range of 1.9–223.3 ng/L.

Graphene nanoribbons (GNRs) are another promising nanomaterial being used in composites with conductive polymers to improve the performance of electrochemical sensors in high-power electrochemical energy storage systems. Reference [74] presents a composite based on polypyrrole and a graphene nanoribbon obtained by the in situ polymerization of pyrrole monomers in an acid medium together with graphene. The composite showed higher specific capacitance, better stability, and better performance than pure pyrrole, making it a promising material for use in supercapacitors. In another study, the synthesis of a composite of polypyrrole–graphene nanoribbons was carried out at the liquid–liquid interface, where pyrrole monomers were dissolved in the organic phase, and the initiator and graphene were dispersed in the aqueous phase [75]. The composite showed high electrical conductivity, good thermal stability, and enhanced electrochemical properties. In [76], theoretical studies of graphene/conducting polymer composites were carried out, as a result of which it was shown that the p-stacking orientation is preferable for such structures. For sandwich complexes of oligopyrrole with graphene nanoribbons (with lower graphene concentrations), interaction energies were found, and excellent additivity was shown. It was shown that the gap between the frontier orbitals (HOMO–LUMO) decreases with increasing graphene concentration. The calculated optical bandgap of the composite based on C_58_H_24_ is about 1.7 eV, which agrees with the declared experimental value (2.1–1.81 eV). The calculated bandgap decreases to about 1.6 eV when the proportion of graphene is increased to C_64_H_26_.

It should be noted that the use of nanomaterials makes it possible to impart conductive properties to initially non-conductive polymers, which expands their application, including in the creation of biosensors, biofuel cells, and supercapacitors. Thus, one of the current topics is the study of the properties and design of various flexible electronic devices based on the graphene–bacterial cellulose combination [77].

Thus, carbon nanomaterials are a promising basis for creating composites for electrochemical devices for energy conversion, storage, and transmission. Methods for covalently bonding conductive polymers to carbon nanomaterials, due to more reliable bonding, will achieve higher device stability than co-electropolymerization, where the polymer is retained due to intermolecular interactions. Therefore, the search for various methods for the covalent bonding of nanomaterials with polymers is most relevant. 

### 3.2. Composites Based on Metal Nanoparticles and Conductive Polymers

Nanomaterials based on metals or metal oxides have mechanical, catalytic, and electrical properties, such as a high surface area and high specific surface activity, fast electron transfer, and good biocompatibility. Such properties are suitable for use in bioanalytical devices. As the size of metal particles decreases, the ratio of molecules/atoms present on the surface increases significantly [78,79]. As a result, Van der Waals forces, electrostatic forces, and magnetic attraction between particles are enhanced. The properties of nanocomposites are determined by the properties of the components, the shape and volume fraction of the filler, the morphology of the system, the nature of the interfacial boundary, and the method of forming the composite [80].

Without proper chemical treatment to reduce the surface energy, nanoparticles very often form clusters or agglomerates that are difficult to individually and uniformly disperse in a polymer matrix. This leads to opaque nanocomposites similar to conventional composites [81]. The choice of technology for the formation of polymer nanocomposites is a key factor in the development of biosensors. The most important methods include the in situ polymerization method, sol–gel synthesis, and electrochemical synthesis.

The principle of the in situ polymerization method is the dispersion of nanoparticles in a monomer solution, followed by the polymerization of the resulting mixture in the presence of an appropriate initiator. In situ polymerization due to the uniform distribution of nanoparticles in the polymer matrix makes it possible to achieve better properties of polymer nanocomposites. This method has found application in the formation of a polyaniline–silver nanohybrid, which was the basis of a sensor for determining the concentration of the pheromone of the bug *Euschistus heros* [82]. The nanocomposite was obtained from a solution containing AgNO_3_ (source of nanoparticles), dodecylbenzenesulfonic acid, and aniline (monomer) after adding an oxidizing agent, ammonium persulfate, to initiate the polymerization of aniline. The nanohybrid sensor had a detection limit of less than 3.1 μg/kg and a lower limit of detectable concentrations of 10.05 μg/kg, and the device opens up a wide range of possibilities for detecting the *E. heros* sex pheromone. This method has not seen wide distribution due to complex sample preparation procedures, expensive reagents, and a limited range of applicable polymers.

A relatively uniform distribution of nanoparticles in a polymer matrix can be achieved by the sol–gel synthesis of a composite. Inorganic salts or highly reactive metal alkoxides are used as starting materials for sol–gel synthesis; the sol is formed as a result of hydrolysis. Heat treatment at low temperature turns the sol into a solid gel. Nanoparticles are covalently linked to a polymer to form a nanocomposite. This method has several limitations. Most of the sol–gel monomers are toxic; it is technologically difficult to prepare crystalline composites based on inorganic oxide nanoparticles, and most of the resulting composites are brittle, which limits the use of the sol–gel method. A nanocomposite based on TiO_2_ and polyvinyl alcohol was obtained by the sol–gel method and used to determine antibiotics in rainwater in [83]. Polyvinyl alcohol was added to the sol obtained from titanium ethoxide. Then, the nanocomposite was subjected to heat treatment in a muffle from 30 °C to 200 °C. The sensor was characterized by a low detection limit, high sensitivity, and a wide linear detection range of the antibiotic ciprofloxacin 0.04 µM, 0.8165 µA/µM, and 10–120 µM, respectively.

Electrochemical synthesis is a simple chemical procedure for obtaining nanocomposites. The advantage of this technology is the possibility of forming nanocomposite films directly on the electrode surface. The authors of [84] showed that the in situ electrodeposition of gold nanoparticles (AuNPs) on the surface of a glassy carbon electrode modified with polyaniline is more efficient with the method of linear sweep voltammetry (LSV) than with the method of cyclic voltammetry (CV). LSV prevents the detachment of gold atoms from the electrode surface, which leads to efficient filling of the polyaniline nanomaterial. The number of nanoparticles deposited by LSV over 15 cycles is 54.75 × 10^−9^ g. It has been shown by IR spectroscopy that the electroreduction of Au increases quinoid fragments in polyaniline, and X-ray diffraction data show that the average size of AuNP nanoparticles is 65 nm. Data from scanning electron microscopy with field-emission and atomic force microscopy indicate the dispersion of spherical nanoparticles over polyaniline. Electrochemical studies show the increased electrocatalytic activity of the modified electrode surface at neutral pH, making it suitable for biomaterial detection and immobilization. The composite was used for the analysis of dopamine with a working linear range of 20–100 µM and a detection limit of 16 µM. This enhanced electrocatalytic response is explained by the synergistic interaction between the conductive polymer and the electrodeposited gold nanoparticles. Thus, the electropolymerization method is the most optimal way to form a nanocomposite for creating biosensors. Electrochemical synthesis is easy to control by changing the applied potential.

Therefore, to date, extensive data have been published on various methods for the synthesis of nanocomposites based on carbon nanomaterials having different sizes and shapes, such as carbon nanotubes, nanowires, fullerene, graphene, graphene oxide, etc. The review discusses nanocomposites with three important conductive polymers (PANI, PPy, PEDOT) in sufficient detail. At the same time, there are few studies of composites based on metal oxide nanoparticles and conductive polymers (especially those containing transition metals), which are widely used as immunosensors, biosensors, gas sensors, and electrochemical sensors and have a good detection limit. Based on the rapid progress in this field and the high-conductivity characteristics of the materials obtained, the mass production of bioanalytical systems based on them is possible in the near future. It should be noted that when nanocomposites are used as parts of devices, improved sensitivity and selectivity with respect to target analytes are noted; however, for the large-scale application of this technology, the issues of the delivery and storage of devices should be considered. Therefore, it is necessary to pay attention to changes in the structure of composites during storage with and without biomaterials, as well as environmental issues arising from the utilization of these multicomponent structures containing nanoparticles, biomaterials, and polymers.

## 4. Biosensors Based on Conductive Polymers

One of the main tasks in the creation of biosensors is to fix the biocatalyst on the electrode surface so that the biocatalyst does not lose its activity for a long time and is also stable or not washed off. As a rule, physical adsorption, covalent attachment, or entrapment in polymer films are the main approaches to biocatalyst immobilization. The choice of strategy depends on the type of biological component. To increase the conductivity, polymers are modified with various nanomaterials. Polyenes and polyaromatic compounds such as polypyrrole, polyaniline, polythiophene, polyterthiophene, poly(3,4-ethylenedioxythiophene), poly(p-phenylene), and poly(phenylenevinylene) are among the most common conductive polymers used to fabricate electrochemical biosensors. Table 2 provides examples of biosensors for the determination of various analytes based on the most commonly used conductive polymers. As can be seen from the table, most polymers are used to create enzyme biosensors. This is due to the practical application of the devices being created. At the same time, microorganisms can initiate the polymerization process, which opens up prospects for obtaining stable coatings on the surfaces of the electrodes. The use of polymers to create affine biosensors is less common, but it is also quite promising, since it allows, on the one hand, the oriented immobilization of a biological component and, on the other hand, the amplification of the biosensor signal and, thus, an increase in the sensitivity of the devices being developed. Next, specific examples of the use of conductive polymers in biosensors of various types are considered.

### 4.1. Enzyme Biosensors

The increased interest in the creation of enzyme biosensors is associated with their further practical applications, mainly in medicine, and particularly with the development of fast, inexpensive methods for assessing the concentration of various blood components that indicate the state of human health. One of these important indicators is the level of glucose in the blood. Therefore, most studies are associated with methods for the immobilization of the enzyme glucose oxidase (GOD), the activity of which can determine the amount of glucose in the sample [85,86].

The transfer of electrons between the active site of the enzyme and the electrode is an important step in the functioning of the biosensor. Thus, one of the topical areas of research is the oriented immobilization of a biocatalyst using conductive polymers, which makes it possible to locate the active site of the enzyme as close as possible to the electrode surface [87]. The simplest method of fixing the bioreceptor on the electrode surface is adsorption. The presence of opposite charges on the conductive polymer and the biomolecule can facilitate the immobilization of the biomolecule. Therefore, negatively charged GOD was successfully adsorbed on positively charged polyaniline–polyisoprene films at pH 4.5, which made it possible to obtain a material sensitive to changes in glucose concentration [88].

The method of covalent immobilization is used for the more stable fixation of enzymes on the electrode. The presence of amino, sulfo, and carboxyl groups allows enzymes to bind to the conductive polymer. In turn, the synthesis of functionalized monomers, followed by electrochemical polymerization, makes it possible to obtain conductive polymer films with the necessary chemical groups for the subsequent covalent binding of enzymes to them. Also, additional crosslinking agents, such as glutaraldehyde, are used to form covalent bonds. For example, a glassy carbon electrode (GCE) was modified with the poly(benzenediamine-bis[(2-ethylhexyl)oxy]benzodithiophene) (P(BDBT)) polymer with functional amino groups and used as an immobilization platform for GOD in [89]. Amino groups available on the polymer backbone served as bioconjugation sites for GOD via glutaraldehyde. The developed biosensor did not respond to the introduction of acetaminophen, ascorbic acid, citric acid, or urea at a concentration of 0.5 mM and was sensitive (28.17 μA/(mM·cm^2^)) to glucose in the range of 0.1–1.0 mM, without losing activity for 20 days.

The capture of an enzyme molecule in a conductive polymer film is another method of bioreceptor immobilization. The surface of the PANI polymer obtained by electrodeposition has a coral-like island structure (Figure 21), which makes it suitable for enzyme immobilization, for example, GOD [90]. The developed biosensor was sensitive to D(+)-glucose in the concentration range of 0.01 M–0.1 M at pH 6.0 and a potential of −0.4 V.

The one-stage immobilization of enzymes during the electropolymerization of a conducting polymer makes it possible to use the entire electrode surface more completely [91]. This fast, simple approach to enzyme immobilization is associated with the formation of polymer films on the surface of the electrode with the simultaneous inclusion of the enzyme in them. During the process, it is also possible to introduce additional reagents into the growing polymer film, for example, coenzymes and/or mediators, as well as nanomaterials to increase conductivity. In this case, the thickness of the formed film and the amount of the immobilized enzyme can be controlled. The most commonly used polymers for this procedure are PPy and PEDOT and their derivatives. For example, in [92], a one-stage procedure for modifying a graphite electrode led to the creation of a composite layer of PPy and GOD on its surface, while the authors simultaneously embedded Prussian blue into the composite. The sensitivity of the biosensor to glucose was in the range of 1.0–1.9 μA cm^−2^ mM^−1^ depending on the initial concentration of pyrrole. Such a system of PPy, graphene oxide, and cholesterol oxidase made it possible to create a highly sensitive (1095.3 μA mM^−1^ cm^−2^) biosensor for the determination of cholesterol with a detection limit of 3.78 µM, with a wide linear range (0.01–6 mM) [93]. The entrapment of the acetylcholinesterase enzyme in a conductive matrix consisting of PEDOT, κ-carrageenan, and gold nanoparticles made it possible to create a highly selective biosensor for the detection of organophosphate pesticides, including chlorpyrifos, in real water samples from rivers and lakes [94].

The use of conducting polymers in the creation of enzyme biosensors makes it possible, in some cases, to achieve direct electron transfer from the active sites of the enzyme to the electrode [19,95]. Direct transfer has been shown for cellobiose dehydrogenase in polyethyleneimine [96], glucose oxidase in polypyrrole [97], laccase in electrostatically condensed oppositely charged polyelectrolyte [98], PQQ-dependent glucose dehydrogenase in PEDOT:PSS [99], and some other enzymes (Table 3). In [100], linear poly(ethyleneimine) (LPEI) covalently modified with pyrene fragments made it possible to create a direct bioelectrochemical interface between a series of redox proteins and a carbon electrode without the need for specific orientation (Figure 22). This method has been applied to promote the direct bioelectrocatalytic reduction of O_2_ by laccase and by immobilizing the catalytic subunit of nitrogenase (MoFe protein) to demonstrate the ATP-independent direct electroenzymatic reduction of N_2_ to NH_3_.

Polymer coatings make it possible to protect the enzyme bioreceptor from the influence of interfering reagents. For example, the first enzyme biosensor based on GOD, created by Clark, contained a polyethylene membrane on the surface of the electrode, impermeable to catalase, which is contained in the blood and affects the oxygen concentration, leading to distortion of the biosensor readings. [101]. Ohara et al. in 1994 deposited a Nafion film on a layer of a redox polymer (based on Os and PEG compounds) and an enzyme (GOD and lactate oxidase), which made it possible to reduce the effects of electrooxidation currents of ascorbate, urate, acetaminophen, and L-cysteine on biosensor signals [102]. The presence of urates in the samples has a negative effect due to the intermediate products of their electrooxidation and can lead to damage to the enzyme layer. The application of an additional Nafion cation-exchange film over the enzyme in the polymer matrix reduces the permeability of the film to the penetration of urates and other compounds. For example, the protective effect of a Nafion film for biosensors based on cholesterol oxidase and bilirubin oxidase immobilized on the conducting polymers PPy and PANI [103] and acetylcholinesterase on microporous organic polymers was shown [104].

To increase the efficiency of bioelectrocatalysis, the electrode is modified with conductive (electroactive) polymers, which contain analogs of substrates in their structure that have an affinity for the active center of the enzyme or for its electron transport chain. For example, GOD and some other oxidases work as oxidizers of various substrates. In addition, under natural conditions, in the presence of dissolved oxygen, oxidases generate H_2_O_2_, which is a strong oxidizing agent and can induce the polymerization of some monomers, such as pyrrole, aniline, phenanthroline, thiophene, and 9,10-phenanthrenquinone. The monomers polymerize and form the corresponding conducting polymers [19,105]. Thus, the catalytic activity of the enzymes is used to induce the biosynthesis of the conductive polymer. In this case, the enzyme self-encapsulates and retains its activity in the biosensor for a long time. The conductive films thus obtained formed the basis of biosensors for the determination of glucose [106,107], catechol [108], lactate [109], and other compounds.

Therefore, conducting polymers are widely used in the formation of enzyme biosensors, performing the function of immobilizing the enzyme, protecting it from interfering reagents, and ensuring the transfer of electrons from active sites to the electrode.

### 4.2. Microbial Biosensors

One of the directions in the development of microbial biosensors is the creation of electrochemically active biofilms (EABs) so that electroactive bacteria are embedded in the matrix of extracellular polymers [110]. The growth rate and density of cells in a biofilm can be controlled by varying the conditions of its formation [111]. The formation of an EAB takes from several days to weeks, and it is quite difficult to obtain an identical EAB [112]. Therefore, artificial EABs are used to encapsulate microorganisms by modifying them with conductive polymers. The authors of [113] shows that the addition of conductive polypyrrole at various concentrations reduces the internal resistance of an artificial biofilm made of sodium alginate, which accelerates the metabolism of the microbe, facilitates extracellular electron transfer, and increases the sensitivity of the biosensor to formaldehyde. However, the transfer of toxic substances in the biofilm was weakened by the addition of PPy, and an excess amount of PPy led to a decrease in the sensitivity of the biosensor. Nevertheless, the developed biosensor can find practical applications and improve the effectiveness of early warnings about water quality.

The use of conductive polymers for the immobilization of microorganism cells as part of a biosensor makes it possible to obtain coatings of biosensor electrodes with biological materials in a simpler and more standardized manner. Conductive polymers for modifying microbial biosensors keep cells on the electrode surface but also perform the function of protecting the biomaterial, which increases the life of the biosensor. For example, in [114], the use of a conductive polymer of PEDOT:PSS/graphene made it possible to achieve a sensitivity to glucose of 22 μA·mM^−1^·cm^−2^, and an additional Nafion coating enabled the achievement of a stable level of signals from bacterial cells for 120 days. The additional modification of the electrode surface with thermally expanded graphite made it possible to obtain electron transfer from membrane fractions of *Gluconobacter oxydans* bacteria through PEDOT:PSS without introducing additional mediators [99].

The doping of bacteria into a PPy film by electrochemical deposition due to the negative charges accumulated on their outer membranes has led to the development of a mediator-type bioelectrochemical BOD sensor using polypyrrole (PPy) immobilized with ferricyanide (FC) as a mediator and *Pseudomonas aeruginosa* for the rapid detection of BOD [115]. The measurement time with this biosensor was 10 min, and the detection limit was 2 mg/L. The polymerization of conductive matrices can be carried out in the presence of microorganism cells, simultaneously trapping them in composite materials on the electrode surface. Thus, alginate and alginate–pyrrole matrices were electrochemically polymerized in the presence of *Chlorella vulgaris* cells to detect p-nitrophenyl phosphate with a biosensor [116]. PANI-PSS with *Brevibacterium ammoniagenes* bacterial cells was used to detect urea [117], and PANI and *Bacillus* sp. were used to detect nitrates [118].

Various microorganisms can themselves initiate polymerization and thus be integrated into the resulting conductive films [119]. For example, the synthesis of polypyrrole by the redox cycle [Fe(CN)_6_]^4−^/[Fe(CN)_6_]^3−^ under the action of metabolic processes occurring in yeast [120] has been shown. To improve the electrochemical properties of cells, the in situ synthesis of polypyrrole (PPy) inside the cell membrane was used to develop a glucose biosensor based on *Aspergillus niger* in [121]. On the one hand, the in situ-formed PPy plays the role of a redox mediator, which promotes the permeability of the cell membrane and facilitates the enzymatic reaction, and on the other hand, it also plays the role of a conductive polymer that improves the contact between the biomolecule and the transducer in the biosensor system. The resulting linear range was limited to 10–50 mM, and the sensitivity increased by up to six times; the LOD value reached 5 mM, and Km decreased to a third of the original value when modifying PPy cells.

The doping of conducting polymers with nanomaterials also leads to an increase in the signals of biosensors based on them and an increase in their sensitivity [122]. The creation of three-dimensional porous composites of graphene materials with polymers makes it possible to create electrodes with an increased effective specific surface area. For example, in [123], *Bacillus subtilis* was immobilized on a three-dimensional (3D) porous graphene–polypyrrole (rGO-PPy) composite, which allowed the cells to remain active for a long time: the signal remained at the initial level for 20 days and then at 86% of the baseline level until day 60 of the experiment; the detection limit was 1.8 mg/L BOD_5_.

Thus, conducting polymers for the immobilization of microorganisms are one of the promising directions in the development of analytical devices. The developed microbial biosensors are mainly used to determine the total toxicity of samples [124,125,126,127] and to analyze the BOD of various water bodies [128,129].

### 4.3. Affinity Biosensors

The possibility of the conjugation of antibodies to conductive polymers opens up broad prospects for the creation of affinity biosensors for the determination of various analytes, such as bacteria, viruses, individual disease marker proteins, etc. [130]. Affinity biosensors, depending on the reaction used, can be divided into two categories: immunosensors based on the antibody–antigen reaction and genosensors based on the use of nucleic acids or aptamers.

As in the case of enzymes, when the electrode surface is modified with conductive polymers, various functional groups are formed on the electrode to bind to biological objects. For example, the electrodeposition of a conductive polymer of poly-2,5-bis(2-thienyl)3,4-diamine-terthiophene (PDATT) results in the appearance of many NH_2_ groups on the electrode surface, which can be used to immobilize antibody molecules through the formation of amide bonds with terminal carboxyl (COOH^−^) groups of the antibody [131]. The authors of [132] present an immunosensor based on this polymer for the quantitative determination of cardiac troponin I (cTnI) up to 0.01 ng/mL in less than 15 min using differential pulse voltammetry (DPV). Figure 23A shows a scheme for the formation of such a biosensor. In this case, the roughness coefficient of the polymer surface after the immobilization of antibodies increased, and the thickness of the surface layer also increased by an order of magnitude. Antibody molecules are attached to the polymer-coated electrode through the formation of amide bonds between the carboxyl group on the Fc region of the antibody and the amino group on the surface of the polymer. Thus, the attached antibody molecules take on a tail orientation, which leads to an increase in the thickness of the surface layer (Figure 23B). After treatment with BSA, the roughness coefficient and the thickness of the electrode surface layer decreased, which may be due to the electrostatic interaction between the immobilized antibody and BSA.

The conjugation of antibodies to conductive Au-PAni nanocomposites through a combination of biotinylated streptavidin makes it possible to properly orient the antibody relative to the electrode surface, which increases the likelihood of Ab-Ag binding, and then, with high specificity in a wide linear range (10^2^–10^6^ copies/mL) with a low limit of detection (121 copies/mL), to detect norovirus in samples (from fecal samples infected with NoV) [133]. The use of conductive polymers for the targeted immobilization of antibodies can significantly improve the analytical characteristics of the developed biosensors [134].

In addition, a conductive polymer can be a catalyst, for example, for the reduction of hydrogen peroxide, and thus not only ensure the fixation of the biomaterial and the conduction of an electron to the electrode but also serve as an indicator of some processes. For example, in [135], the use of a conductive polymer composite of phaseoloidin-doped poly(3,4-ethyloxythiophene) (PL/PEDOT) made it possible to obtain an electrode with a large surface area, high conductivity, and high stability, which was then functionalized with a poly(amidoamine) dendrimer of the third generation (PAMAM) and immobilized aptamers. At the same time, phaseoloidin was a source of a large number of functional groups (–COOH, –OH) for the immobilization of biomolecules. A biosensor was made to detect the pesticide acetamiprid (ACE). The concentration of this pesticide could be measured by both the change in the differential pulse voltammetry (DPV) signal and the change in the chronoamperometry (CA) signal as a result of hydrogen peroxide reduction catalyzed by PL/PEDOT. The specific binding of the pesticide by the aptasensor prevented the transfer of electrons to the electrode surface, which led to a decrease in the electrochemical signal and a decrease in the catalytic reduction ability of H_2_O_2_. The PAMAM/PL/PEDOT-based ACE aptasensor exhibited linear ranges of 0.1 pg/mL–10 ng/mL and 0.1 fg–1.0 pg/mL, and the detection limits were 0.0117 pg/mL and 0.0355 fg/mL (S/N = 3) through DPV and CA, respectively. Notably, the response sensitivity of DNA/PAMAM/PL/PEDOT/GCE was 3.11 times higher than that of DNA/PL/PEDOT/GCE (in CA mode), proving that the PAMAM dendrimer can significantly improve the recognition performance.

Polymer composites containing nanoparticles of various natures are used for practical applications. Nanomaterials facilitate the immobilization of biological material, enhance the signal, obtain higher sensitivity and selectivity of a bioelectrochemical device, and determine lower levels of the analyte [136]. Accordingly, the developed devices find wider practical applications, especially in medicine, where it is necessary to control various health indicators with high accuracy, especially in the diagnosis of cancers, for example, breast cancer [137] and neurotransmitters [138]. For example, an electrochemical sensor based on a conductive polymer composite with a complex of metal palladium nanoparticles (Pd(C_2_H_4_N_2_S_2_)_2_) has been developed for the simultaneous detection of serotonin and dopamine in samples of breast cancer cells and human plasma [139]. The proposed sensor was fabricated using a Pd(C_2_H_4_N_2_S_2_)_2_ complex-anchored poly2.2 : 5,2-terthiophene-3-(p-benzoic acid) (pTBA) layer on AuNP-decorated reduced graphene oxide (AuNPs@rGO) substrate, which revealed the enhanced anodic current of the target species. The authors of [140] present a biosensor based on carbon fiber microelectrodes with electrodeposited PEDOT and graphene oxide nanoparticles for the determination of the monoamine neurotransmitter dopamine with a lower detection limit of 0.085 μM. The dependence of the adsorption kinetics, electron transfer in the device, sensitivity, and detection limit of the analyte on the thickness of the electrodeposited composite was studied.

The poly(glutamic acid) (PGA) polymer contains a large number of protonated carboxyl groups in the side chain, which can bind to biomolecules through electrostatic and covalent interactions. Accordingly, the use of PGA in combination with nanomaterials, for example, nanotubes, leads to an increase in the area of binding of biological recognition molecules, conductivity, and, as a consequence, sensitivity [141]. The synergistic interaction of the components of the composite leads to an increase in the sensitivity and selectivity of the biosensor, such as an aptasensor based on a ternary nanocomposite of reduced graphene oxide, chitosan, and gold nanoparticles, as presented in [142]. Due to the presence of RGO and AuNP in the nanocomposite, the charge-transfer characteristics improved, and due to this, the aptasensor showed a linear relationship with the cell concentration logarithm, high selectivity, a wide linear range of 1 × 10^1^–1 × 10^6^ cells/mL, and a low detection limit of 4 cells/mL of MCF-7 cancer cells.

In addition, the use of nanomaterials in combination with polymers makes it possible to create three-dimensional electrodes with an increased concentration of the biological recognition component due to an increase in the active surface area. For example, using a three-dimensional composite of PEDOT, Prussian blue, and gold nanoparticles, an immunosensor for the label-free detection of carcinoembryonic antigen (CEA) was fabricated in the range of 0.05 to 40 ng/mL, and the limit of detection was 0.01 ng/mL in [143]. The 3D nanocomposite with a porous structure had a large surface area and redox properties and also created a favorable microenvironment for antibodies, demonstrating remarkable conductivity, stability, and biocompatibility of the final device.

Therefore, in affinity sensors, on the one hand, conductive polymers serve as a matrix for immobilizing antibodies, receptors, or other compounds capable of selectively binding an analyte. On the other hand, conductive polymers can be an indicator of changes in the semiconductor properties of the polymer as a result of interaction with the analyte. 

As can be seen from the examples, conductive polymers are an important component of developed biosensors. It should be noted that the use of conductive polymers, especially modified nanomaterials, not only allows the improvement of the analytical parameters of biosensors but also leads to the miniaturization of the devices being developed. In addition, in some cases, the specific binding of the bioreceptor to the analyte can be detected without the use of additional labels or mediators. Understanding the processes of the emergence and transmission of signals in such structures is one of the leading directions in the development of conductive polymers and devices based on them. Additional research will allow not only the study of the mechanisms of signal transmission in systems based on conductive polymers but also the production of biosensors with high selectivity. One of the important tasks in the development of such biosensors remains to ensure the stability and uniformity of methods for manufacturing conductive electrode coatings, since, currently, most of the work is related to laboratory research and does not reach the level of production scales. From here, it can be quite difficult to compare the parameters of the coatings obtained. The study of methods for including biomolecules in electrodeposited conductive polymer films will allow the bioreceptor to be immobilized on electrodes of any size and shape and will allow the creation of biosensors for the simultaneous determination of several analytes in samples of complex composition. The use of conductive polymers in biosensors allows the adjustment of the coating thickness, chemical composition, and physical and mechanical properties of the resulting films. At the same time, these processes are still insufficiently studied and require optimization. In addition, measurements by such biosensors of real samples with a high content of metal compounds can be difficult, since the electrical conductivity of conducting polymers changes by several orders of magnitude in response to changes in pH, oxidation–reduction potential, or the environment. Thus, in order to increase the sensitivity, selectivity, and speed of analysis, it is necessary to study such parameters as the pH and ionic strength of solutions in which polymers are formed and analytes are measured with the final biosensor. Studies of systems combining conductive polymers and non-organic materials, especially carbon ones, also have great potential. And an important direction also remains the search for increasing the long-term stability of biosensors based on conductive polymers, which will allow continuous measurements over long periods of time.

**Table 2 polymers-15-03783-t002:** Examples of composite biosensors based on commonly used conductive polymers for the determination of various compounds.

Polymer	PolymerSynthesis Method	Composite	BiosensorFormation Time	Determined Compound	Real Samples	Biosensor Type	Detection Method	Main Specifications/Detection Limit/Detection Range	Reference
Polypyrrole 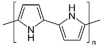	One-step electrochemical deposition	PPy/polydopamine/(GOx)	1 h, holding 1 day	Glucose	Human blood serum	Enzymatic (GOX)	CA, CV, EIS	Sensitivity—22.15 A mM^−1^ cm^−2^, response time—5–6 s, linear range up to 5.0 mM; LoD—138 µM glucose; stability for 90 days (93.9%).	[144]
	Electropolymerization	NAD- GDH/poly-TBO (Poly-toluidine blue)/Ppy/SPE	14 h	Glucose	Synthetic urine	Enzymatic (GDH)	CV, EIS, CA	Linear range—1.0 × 10^−3^ to 9.0 × 10^−3^ M; LoD—9.0 × 10^−5^ M	[145]
	One-step electrochemical copolymerization of pyrrole (PPy) and chondroitin sulfate (CS)	CS/PPy nanowires	Approximately 2 h	Acetamiprid (insecticide)	Soil samples	Aptasensor	CC, CA, EIS	The determination time—0.5 s and 2 s; LoD—0.347 pg/mL and 0.065 fg/mL	[146]
	Electrochemical polymerization	Polyethylene glycol (PEG)/PPy nanowires	1 h	MicroRNAs (miRNAs)	serum samples	DNA probes	DPV	Linear range—0.10 pM ∼ 1.0 nM, LoD—0.033 pM; RSD—3.05%	[147]
Polyaniline 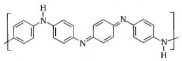	Electrophoretic deposition	AChE/Ag@CuO/PANI/ITO	14 h, preliminary procedures for several days	Paraoxon-ethyl	Banana, tomato, and soil	Enzymatic (AChE)	CV, EIS	Linear range—5–100 pM; LoD—11.35 pM; sensitivity—0.5536 μA pM^−1^ cm^−2^; RSD—1.74%; After 20 days of storage, the current response remains 71.3% of its initial current	[148]
	Electropolymerization	Nf/PANI/CuF/Urease	3 days	Urea	Soil and milk samples	Enzymatic (urease)	CV, DPV	LoD—0.17 µM; linear range—0.5–45.0 µM	[149]
	Electropolymerization	Phytic acid/PANI/SCoV2-rS	1.5 h	Antibodies against severe acute respiratory syndrome coronavirus 2 (SARS-CoV-2) Spike protein	-	recombinant Spike protein (SCoV2-rS)	EIS	LoD—8.00 nM; range to 23.93 nM	[150]
	Electropolymerization	Olyaniline titanium oxide (PANI-TiO_2_)/monoclonal antibodies specific to l-glutamic acid	14 h	L-glutamic acid	Tomato sauce	Immunosensor (anti-glutamate monoclonal antibodies)	DPV	Sensitivity ~37 mA/nM; detection ranges 1 nM to 500 µM in the electrolyte, 1 µM to 250 µM in tomato sauce	[151]
Polythiophene 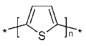	Electrochemical polymerization	Polythiophene film/graphene oxide (GrO)/GOx	13 h	Glucose	Commercial fruit juice samples (pear and apricot)	Enzymatic (GOx)	CA	Linear range—0.2–10.0 mM; LoD—0.036 mM; sensitivity—9.4 µA mM^−1^ cm^−2^; Response Time—10–20 s; Stability >60 Days	[152]
	Amperometric depositPon	AuNPs- poly(thiophene-3-boronic acid) (PT3BA)- tyrosinase enzyme	-	Dopamine	human urine sample	Enzymatic (tyrosinase)	DPV, CV	Linear range of detection—5 × 10^−8^–3 × 10^−5^ M; LoD—2 × 10^−8^ M; RSD—3.2%; The lifetime was at least 2 months (89%)	[153]
	Electrochemical polymerization	(Poly)thiophenes, namely 2,2′-bithiophene (poly(2,2′-BT))/GOx	1 day	Glucose	Fruit juices (pear, peach, and apricot)	Enzymatic (GOx)	CA, CV	LoD—30 μM; Linear range of detection—0.09–5.20 mM; response time—120–180 s; stability >15 days	[154]
		4,4′-bis(2-methyl-3-butyn-2-ol)-2,2′-bithiophene (poly(4,4′-bBT))/GOx						LoD—50 μM; Linear range of detection—0.15–5.20 mM; response time 20–50 s; stability >30 days	
PEDOT 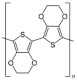	Electrodeposition	Au-XOR/fMWCNT-PEDOT/GCE	6 h	Xanthine	Human serum and urine samples; fish and meat samples	Enzymatic (xanthine oxidoreductase (XOR))	DPV, CV, EIS	LoD—5.45 × 10^−2^ мкM; Linear range—0.1–10 мкM; response time—4 s, sensitivity—16.075 µA.µM^−1^cm^−2^), stability—4 months	[155]
	Deposition	*G. oxydans*/PEDOT: PSS/graphene/Nafion	1 day	Glucose	-	Microbial (*G. oxydans)*	CV, EIS, CA	Sensitivity—22 μA × mM^−1^ × cm^−2^; concentration range 0.02–2 mM; LoD—0.02 mM; Stability >120 Days	[114]
	Deposition	Metal-organic framework (MOF), i.e., MIL-53 (Fe) (MIL = Materials of Institut Lavoisier)/PEDOT:PSS/*anti-E*. coli antibodies	Several days	*E. coli*	-	Immunosen-sor	DPV, EIS	Concentration range—2.1 × 10^2^ –2.1 × 10^8^ cfu/mL, LoD—4 cfu/mL	[156]
Poly(p-phenylene) 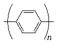	Electrodeposition	Poly(p-phenelyne) modified in side chain position with ferrocenyl group (Fc-PPP)/β-alanin/DNA	-	DNA	-	DNA	CV, EIS,	LoD—30 fM; range of detection to 10 pM; electron transfer kinetics with a value of 68 s^−1^	[157]

List of abbreviations: Acetylcholinesterase (AChE); Chondroitin sulfate (CS); Chronoamperometry (CA); Chronocoulometry (CC); Cyclic voltammetry (CV); Deoxyribonucleic acid (DNA); Differential pulse voltammetry (DPV); Electrochemical Impedance Spectroscopy (EIS); functional multiwalled carbon nanotubes (fMWCNT); Glassy carbon electrode (GCE); Glucose dehydrogenase (GDH); Glucose oxidase (GOx); Graphene oxide (GrO); Indium Tin Oxide (ITO); Poly(3,4-ethylenedioxythiophene) (PEDOT); Poly(3,4-ethylenedioxythiophene) polystyrene sulfonate (PEDOT:PSS); Polyaniline (PANI); Polyethylene glycol (PEG); Pyrrole (PPy); Relative standard deviation (RSD); Ribonucleic acid (RNA); Screen-printed electrode (SPE); The limit of detection (LoD); β-nicotinamide adenine dinucleotide hydrate (NAD).

**Table 3 polymers-15-03783-t003:** Analytical parameters of biosensors based on direct electron transfer containing conductive polymers.

Conductive Polymer	Composite on the Electrode Surface	Enzyme	Electrode Material	Determined Compound	Analytical Parameters	Reference
PEI	PEI@AuNP	CDH	Gold disk electrode	Lactose	Electron-transfer (ET) rate (39.6 ± 2.5) s^−1^; linea range from 1 to 100 μm; response time less than 5 s	[96]
PPy	Nafion-GOx-fMWCNTs-PPy	GOx	Pt electrode	Glucose	Sensitivity (54.2 μA mM^−1^ cm^−2^) in the linear range up to 4.1 mM, LoD—5 μM, response time within 4 s,	[97]
PEDOT	(PAN-MWCNTs)/PEDOT	GOx	Pt disk	Glucose	Sensitivity 92.94 µA/mM cm^−2^; LoD—2.30 µM; linear range 0.01–1.2 mM	[158]
PPy	PAN-MWCNTs)/PPy	GOx	Pt disk	Glucose	Sensitivity 81.72 µA/mM cm^−2^; LoD—2.38 µM; linear range 0.01–2.0 mM	
PEDOT:PSS	GP-PEDOT:PSS	GOx	SPE	Glucose	Sensitivity of 7.23 μA/mM; linear range of 20–900 μM; LoD—0.3 μM; enzyme activity decreases by 30% after 30 days.	[159]
PANI	PANI/SnO_2_-NFs	Catalase	GCE	H_2_O_2_	Linear range 10 to 120 μM; LoD—0.6 μM; stability 92% (35 days)	[160]
PPy	PPy-Cl-PPy	GOx	Pt-disc	Glucose	Linear range of 0.5–24 mM and LoD—26.9 μM; Highly stable reponse for more than 2 months; sensitivity 3.5 μA cm^−2^ mM^−1^; 1.9% RSD; Rejects interferences from ascorbic acid, glycine, glutamic acid and uric acid	[161]
DTP(aryl)aniline	GCE/p DTP(aryl)aniline/ChOx	ChOx	GCE	Cholesterol	Linear range 2.0 μM–23.7 μM; LoD—0.27 μM; sensitivity of 11,246 μA/μM; biosensor lost its 45% of initial activity after 25 days.	[162]
PPI dendrimer	GCE/PPI/QDs/ChOx	ChOx	GCE	Cholesterol	Linear Range 0.1–10 mM; LoD—0.075 mM; Sensitivity 111.16 μA mM^−1^ cm^−2^. After a month of storage at 4 °C, the biosensor retained 97% of the original response in the same sample.	[163]

List of abbreviations: 4-(4H-dithienol[3,2-b : 2′,3′-d]pyrrole-4)aniline polymer (DTP(aryl)aniline); Cellobiose dehydrogenase (CDH); Cholesterol oxidase (ChOx); functional multiwalled carbon nanotubes (fMWCNT); Glassy carbon electrode (GCE); Glucose oxidase (GOx); Graphen (GP); Nanofibers (NFs); Platinum (Pt); Polyacrylonitrile (PAN); Poly (propylene imine) (PPI); Poly(3,4-ethylenedioxythiophene) (PEDOT); Poly(3,4-ethylenedioxythiophene) polystyrene sulfonate (PEDOT:PSS); Polyaniline (PANI); Polyethylene imine (PEI); Pyrrole (PPy); quantum dot (QDs); Screen-printed electrode (SPE).

## 5. BFCs Based on Conductive Polymers

The use of conductive polymers and films based on them makes it possible to immobilize biological molecules on the surface of electrodes of biofuel cells of any shape and size. These devices are used to convert the chemical energy of organic or inorganic substances into electrical energy using biochemical reactions [164]. Basically, either enzymes immobilized on the anode (enzymatic fuel cells) [165] or microorganisms and their consortia (microbial fuel cells) [166,167,168,169] are used as biological objects in biological fuel cells. In addition, some enzymes (for example, laccases) or microorganisms can be used as part of a fuel cell biocathode [170,171,172]. In each of these cases, conductive polymers can be used as the base or doping material for BFC electrodes, since they have all the necessary properties—high conductivity, biocompatibility, and sufficient resistance to external environmental conditions [4]. In addition, in some BFCs, polymer-based membrane separators are used to separate the anode and cathode compartments [173], which are also important parts of the biofuel cell and require separate consideration. Polymer electrolyte membranes play a key role in increasing the efficiency of the BFC by reducing the possible diffusion of the substrate and oxygen from the anode to the cathode compartment and vice versa. However, the high resistance of these membranes may increase the overall resistance of BFCs [174]. In most cases, Nafion-type perfluorosulfonic proton exchange polymeric membranes [175,176] are used as a partition, which have good stability under various operating conditions and have become something of a standard in the creation of BFCs [177]. As a cheaper alternative, heterogeneous membranes made of polypropylene, polyethersulfone [178], polybenzimidazole [179], and polyvinylpyrrolidone [180] are used.

Despite the important role of separating membranes, the most commonly used conductive polymers are still used as substrates (or as part of immobilization composites) of anode and cathode biocatalysts. As part of the anode, they provide a high surface area of the electrode for the immobilization of bacteria and the necessary functional groups for the covalent binding of enzymes. The use of polymers in the composition of the cathode can improve the power densities for both abiotic cathodes and biocathodes [181]. In the first case, the use of polymers improves the oxygen reduction reaction (ORR) activity of the electrode. In the second case, it creates better attachment conditions for the biocatalysts used on the cathode (laccases or microbes). Table 4 provides examples of biofuel cell anodes incorporating conductive polymers. Further, specific examples of the use of conductive polymers in the anodes and cathodes of enzymatic and microbial BFCs are considered.

### 5.1. Enzymatic Anodes

Similar to other fuel cells, enzymatic fuel cells (EFCs) consist of a two-electrode cell separated by a proton-conducting medium. Redox enzymes are used as catalysts, which oxidize the fuels on the surface of the anode, and the electrodes flow through the external circuit to the biocathode. The advantages of this type of fuel cell include the renewability of the biocatalyst, a wide range of possible fuels, and environmental friendliness. The disadvantages are a low energy density and power density, poor operational stability, and limited voltage output [182]. The use of new conductive polymers can both solve problems with the low operational stability of the biocatalyst and improve its electrical properties. Nanomaterials such as metal nanoparticles, carbon nanomaterials, etc., are often used to improve the conductive properties of polymers in bioanodes. However, their use complicates and increases the cost of the fuel cell design, so it is desirable to use polymers that are themselves highly conductive. However, a combination of conductive polymers and conductive nanomaterials is often used to achieve maximum performance [19]. Another desirable property of polymers used in EFCs is their biocompatibility, as BFCs of this type are most often used as wearable and implantable fuel cells [183,184].

One of the most commonly used conductive polymers in EFCs is poly(3,4-ethylenedioxythiophene) (PEDOT), which has seen broad adoption in biological–electronic interfaces [185]. Most often, a mixture of two monomers, PEDOT and poly(styrenesulfonate) (PSS), is used, which makes it possible to stabilize the structure of the resulting PEDOT:PSS polymer [186,187]. PEDOT:PSS is often further modified with carbon nanomaterials to stabilize the polymer, improve electrical conductivity, and increase the bioanode surface area [188,189]. For example, Cho et al. [190] were the first to describe a paper-based wearable EFC capable of measuring glucose levels in sweat. The anode was modified with a mixture of PEDOT:PSS and graphene nanoparticles containing glucose. The wearable BFC provided a biosensor that showed a good sensitivity to glucose within the dynamic range of 0.02–1.0 mg glucose mL^−1^. PEDOT:PSS can also be used in hydrogels used as self-standing MFC bioanodes. In [191], a three-component electroconductive hydrogel containing polyvinyl alcohol, carrageenan, and PEDOT:PSS became the basis of an EFC electrode integrated into a stent. The high biocompatibility of the obtained material will make it possible to use it as part of BFCs implanted into the cardiovascular system. Another important feature of PEDOT is its redox activity, due to which EFCs based on direct electron transfer can be developed. In particular, Kwon et al. [192] demonstrated a biofuel cell with a maximum power density of 236 mW cm^−2^ using a self-assembled yarn electrode consisting of multi-walled carbon nanotubes, PEDOT, and two enzymes without the use of additional mediators.

PEDOT is not the only conductive polymer used in fuel cells. However, most other polymers require pretreatment or the incorporation of nanomaterials to improve their conductive properties. In particular, the organic polymer polypyrrole (PPy) is often used in EFC anodes [193]. PPy itself is an insulator, but its oxidized derivatives are good conductors. In [194], Kang et al. proposed the carbonization of rectangular polypyrrole at high temperatures. The resulting electrode was used to immobilize GOx and laccase and as part of an enzymatic fuel cell that provided a maximum open-circuit voltage of 1.16 V. Kizling et al. [195] showed that a pseudocapacitive cellulose/polypyrrole composite improved the performance of an enzymatic fuel cell based on fructose dehydrogenase. The introduction of a carboxylic group into the composite improved the stability of the electrode and enabled more efficient FDH adsorption, due to which the developed electrodes exhibited an OCV of 0.6 V, a maximum generated current density of 7.7 mA cm^−2^, and a maximum power density of 2.1 mW cm^−2^ under a load of 1 kΩ. Polyaniline (PANI) is interesting in that it allows the creation of porous matrices for the simultaneous entrapment of redox mediators and enzymes on the surfaces of bioanodes. At the same time, it has good stability, it is cheap to manufacture, and its conductive properties can be changed during synthesis. The direct contact of all components of the bioanode provided by PANI makes it possible to achieve an enhanced electrochemical signal transfer rate from the enzyme to the electrode surface [196,197]. In addition, PANI is used to create 3D composites and 3D electrodes that can be used as bioanodes. A 3D composite of carbonized PANI and CNT forms a rhizobium-like structure containing glucose oxidase and laccase on the surfaces of glassy carbon electrodes [198]. Glucose//O_2_ EBFC, consisting of an anode and a cathode fabricated by this technique, provides a maximum power density of 1.12 mW cm^−2^ at 0.45 V. Three-dimensional Ni-foam electrodes were coated with aminated graphene and composites for an EFC based on glucose [199]. At the same time, the polymerization of aniline in the presence of aminated graphene under different conditions made it possible to obtain composites with different properties. The maximum power density was observed to be 118 μW cm^−2^. Recently, EFC creators have turned their attention to commercially available synthetic polymers, usually employed in fuel cell and flow battery applications. In particular, a commercial conductive polymer (PV15) was used as the basis for a bioanode with immobilized GOx and a biocathode with laccase [200]. The resulting cell was able to generate power for three weeks, reaching 590 mV and 2.41 µW cm^−2^ as the maximum open-circuit voltage and power density, respectively.

Therefore, conducting polymers are often used as part of enzyme bioanodes due to their ability to rapidly transfer electrons (due to the delocalization of π electrons in the polymer structure), their high biocompatibility, and the ability to create complex multicomponent matrices with controllable properties based on them by varying the synthesis conditions or polymer modifications. In the vast majority of cases, even well-conducting polymers are additionally modified with either carbon nanomaterials [201] or metal nanoparticles [202] to further accelerate electron transfer from the active redox site of enzymes to the anode surface.

### 5.2. Microbial Anodes

The performance of microbial fuel cells mainly depends on the conductivity of the electrode materials, the reactor configuration, and the biocatalysts used. The rate of extracellular electron transfer between the microbial layer and the anode surface is considered to be the most important factor for BFC operation [203]. At the same time, the limited surface area of the working electrodes, as well as insufficiently tight contact between the biocatalyst and the anode, can be attributed to the factors limiting the performance of the BFC. Therefore, the modification of BFC anodes is mainly aimed at increasing the specific surface area and improving the conductive properties of the surface [204]. Both of these problems can be solved by using suitable conductive polymers in the composition of the bioanode. In addition, the use of conductive polymers is necessary to improve the process of the attachment of bacterial cells to the electrode [205].

Recently, many studies have been devoted to the modification of anodes with semiconductor polymers, such as polyaniline (PANI), polydopamine (PDA), polypyrrole (PPy), and PEDOT [206]. The work of researchers has been aimed at both the use of conductive polymers for the immobilization of pure cultures on bioanodes (mainly bacteria of the genera *Geobacter* [207] and *Shewanella* [208]) and the fixation of multicomponent microbial communities on the anode [209]. The use of conductive polymers in the BFC composition has a positive effect on the total number of cells in the emerging biofilm and also reduces the time of its formation. For example, it was shown in [210] that the use of PANI modified with graphene made it possible to reduce the acclimatization time for a biofilm by 2.4 compared to a pure graphite anode upon inoculation with wastewater. A positive effect of conductive polymers on the electrogenic properties of bacteria was also shown, which directly affects the power generated by the BFC. Differences in the power generated by BFC mock-ups based on graphite electrodes coated with PANI, rGO, and carbon nanotubes (CNTs) were studied [211]. It was the electrode coated with PANI (324.2 mA cm^−2^), followed by CNTs (248.75 mA cm^−2^), rGO (193 mA cm^−2^), and blank (without coating) (151 mA cm^−2^) graphite electrodes, that could develop the maximum power. At the same time, it was shown that the choice of bioanode coating has a significant impact on microbial diversity. The biofilm harvested from a PANI-coated bioanode had significantly more observed operational taxonomic units than other materials.

In recent years, the environmentally friendly material polydopamine (PDA) has attracted much attention due to its ease of use and high hydrophilicity. In [212], it was shown for the first time that the use of 50% PDA as part of the BFC bioanode made it possible to increase the MFC power density by 31%. The Coulombic efficiency of the BFC in municipal wastewater treatment also increased from 19% to 48%. PDA is also used in the creation of 3D bioanodes; in particular, a composite 3D anode made with a combination of silk fibroin and polydopamine with a carbon base is presented in [213]. A high power density of ∼9 W/m^2^ was obtained with an ∼84% COD removal efficiency. At the same time, PDA improves the characteristics not only for BFCs based on bacterial consortia but also on the basis of pure cell cultures. Thus, the use of PDA for coating *Shewanella xiamenensis* cells made it possible to create a BFC generating a specific power of 452.8 mW m^−2^, which is 6.1 times higher than that of electrodes based on unmodified cells (74.7 mW m^−2^) [214].

Polypyrrole is also actively used to create MFC bioanodes, although there are reports of its cytotoxic effect, which significantly hinders metabolic activity in yeast [215]. Nevertheless, the advantages of PPy include its redox activity, tunable conductivity, and reversible and low-cost synthesis [216,217]. For example, in [218], a bioanode made of polypyrrole, carboxymethyl cellulose, and carbon nanotubes/carbon brushes is presented. The bioanode demonstrated a fourfold improvement in the power of the BFC mock-up compared to the bare anode. Polypyrrole in combination with Fe_3_O_4_ magnetic nanoparticles ensures the effectiveness of sewage treatment. Fan et al. [219] used Fe_3_O_4_-PPy as a bioanode, and the chemical oxygen demand (COD) removal rate was 95.3% higher than that of the unmodified anode. PPy can simultaneously perform the functions of a capacitive material and a conductive intercalator between the bacterial layers when using the layer-by-layer technique to create bacterial bioanodes. Thus, in [220], this technique was applied to create a bioanode with improved electron-transfer efficiency, enhanced power density, and increased capacitance. Polypyrrole not only contributed to the immobilization of bacteria to improve traditional BFC functions but also had a protective effect on bacterial layers, leading to a superior performance-retention capacity during the storage period when using this bioanode as a supercapacitor electrode. In the future, the use of conductive polymers may increase the level of integration between BFC and supercapacitor technologies and lead to the development of new hybrid multifunctional devices [221].

PEDOT has superior conductivity compared to PPy, and PANI [222] also has potential as an excellent electrode modifier in microbial fuel cells. PEDOT has been used, for example, in combination with nickel ferrite (NiFe_2_O_4_) nanorods and biochar to create a free-standing and binder-free anode in microbial fuel cells [223] and also in combination with nickel nanoparticles and thermally reduced graphene [224]. The addition of PSS (polystyrene sulfonate), acting as a surfactant and doping agent, allows polymerization to be carried out at higher monomer concentrations in solution; in addition, the use of PSS improves the electron transfer in the composite. The authors of [225] demonstrated a new electrode modification process via the in situ electropolymerization of PEDOT doped with PSS for use as anodes in urine-fed MFCs. The authors showed that anodes made of this material provided sufficient stability for MFCs (90 days) and power 24.3% higher than when using bare carbon veil electrodes. One of the disadvantages of PEDOT is the demonstrated antibacterial effect [226], as well as the low pH of the composite solution, which can adversely affect the lifetime of the microbial bioanode. To eliminate limitations, the authors of [227] used the layer-by-layer deposition of a PEDOT:PSS/graphene/Nafion composite in combination with the acetic acid bacteria *Gluconobacter oxydans*, which are sufficiently resistant to the acidic properties of PEDOT:PSS and Nafion. As a result, a pure-culture microbial BFC was developed that can effectively treat municipal wastewater and shows potential when used in combination with a DC-DC boost converter as part of the Internet of Things technology. PEDOT:PSS, as part of a BFC, is used not only in combination with graphene but also with thermally expanded graphite [228]. The PEDOT:PSS/TEG@CF anode demonstrated higher ion transport capability, superior bioelectrochemical conductivity, and excellent capacitance compared to standard carbon anodes. Also, not only carbon but also metal electrode materials can be used as a basis for PEDOT:PSS-based bioanodes. For example, Shetty et al. [229] used nickel foam modified with magnesium cobalt oxide and PEDOT:PSS. The maximum power and current density values were found to be 494 mWm^−2^ and 900 mA m^−2^ using MgCoO_2_/PEDOT:PSS@NF anode, because the nickel foam provided the necessary porosity of the bioelectrode for the stable formation of a biofilm from exoelectrogenic microbes present in wastewater, and PEDOT:PSS ensured efficient electron transfer.

In general, the field of application of conductive polymers for the immobilization of microbial cells has been rapidly developing in recent years. Further efforts of researchers will be focused mainly on the synthesis of new composites based on conductive polymers and nanomaterials, as well as on the possibility of using the developed composites in various bioelectrochemical systems, including MFCs, microbial solar cells, microbial electrolysis cells, etc. [230,231].

### 5.3. Cathodes

Cathodes play an important role in determining the performance of biofuel cells, especially parameters such as their power density and cost [182,232]. Various types of cathodes are used in fuel cells: air cathodes, water–air cathodes, and biocathodes [233]. They can also be modified with additional catalysts, both biological and non-biological. The main problem in the manufacture of an efficient cathode electrode for a BFC is the preservation of its highly conductive and catalytically active surface for a long period of operation. Thus, it is believed that the cost of the cathode, on average, is up to 50% of the total cost of the entire biofuel cell [234]. Therefore, optimizing the characteristics of the cathode is one of the most important tasks in the development of BFCs. Conductive polymers can be one of the solutions that will accelerate the interfacial electrochemical reaction on BFC cathodes and bring them closer to practical applications in various fields. Table 5 provides examples of biofuel cell cathodes incorporating conductive polymers. One of the first polymeric materials used in BFC cathodes was PANI. In the work of Li et al. [181], PANI and three of its copolymers with different functional groups were used to modify abiotic cathodes and biocathodes, and their effects on oxygen reduction ability and biofilm characteristics were studied. All studied materials acted as favorable biofilm supports and greatly enhanced the power densities of MFC mock-ups created on their basis. In this case, modification of the polymer with –OH groups led to its lower pH sensitivity, and modification with –NH_3_ groups made the cathode more capable of biomass adhesion. There are also works devoted to the incorporation of nanomaterials (for example, graphene and TiO_2_ nanoparticles) into a PANI matrix to accelerate the oxygen reduction reaction at the cathode [235]. The developed PANI-TiO_2_-GN nanocomposite was simultaneously used both as a bioanode component and as part of the MFC cathode, which generated a high power density of 79.3 mW/m^2^.

In microbial BFC cathodes, electrically conductive polymers are also used as a replacement for standard platinum/graphite cathodes to improve oxygen reduction reaction (ORR) kinetics. In particular, Polyindole together with iron phthalocyanine was immobilized on carbon nanotubes and on Vulcan carbon. The resulting cathode showed better electrocatalytic activity and higher kinetic current density values compared to those of the conventional ones [236]. Polypyrrole was also used as a component of the inorganic cathode. Sumisha and Haribabu modified carbon cloth using polypyrrole nanoparticles for the cathode in a single-chamber membrane-less microbial fuel cell for bioenergy production and iron removal [237]. And in [238], MFCs employing the Ni–NiO/PPy–rGO nanohybrid catalyst exhibited a maximum current density of 2134.56 mA m^−2^ and a maximum power density of ~678.79 ± 34 mW m^−2^, which is 25–30 percent higher than when using a commercial Pt/C catalyst.

Conductive polymers are also used to modify metal–organic frameworks, which, in recent years, have often been used as BFC cathodes due to their high activity as ORR catalysts. In particular, in [239], the MIL-53 metal–organic framework was modified with a conductive PEDOT gel. The resulting cathode exhibited a four-electron-transfer pathway, which helped the developed MFC achieve a power production of 4.78 W/m^3^ in a ferricyanide catholyte. At the same time, the cost of the developed BFC was ~6.7 times lower than that of the benchmark with a Pt/C cathode. Finally, Simeon et al. [240] compared four polymeric materials (polytetrafluoroethylene, two-component epoxy, polyvinyl alcohol, and polyvinylidene fluoride) when used as a binder for a stainless-steel anode and cathode in a soil microbial fuel cell. At the same time, the polyvinylidene fluoride-modified electrode showed the lowest charge-transfer resistance (3 ohms), due to which the PVDF-based BFC generated the highest current of 92.2 mA/m^2^ during long-term operation with an external load of 1 kΩ compared to epoxy, PTFE, and PVA, which generated 69.3, 28.9, and 14.5 mA/m^2^, respectively. Therefore, the high conductivity of PVDF has a positive effect on both the performance of the anode populated with soil bacteria and the performance of the abiotic cathode.

If we talk about the use of conductive polymers in the biocathodes of biofuel cells, then their primary role is not too different from the role played by polymers in the modification of bioanodes. However, a different set of biocatalysts is often used in biocathodes than in bioanodes; therefore, some features of the use of conductive polymers may differ here [241]. Microbial biocathodes attract the attention of researchers due to their low cost, the ability of the catalyst to self-regenerate, and their sufficient stability. In this case, a wide variety of aerobic and anaerobic bacteria, cyanobacteria, and microalgae are used as biocatalysts for cathodes [242,243]. In addition, microbial metabolism in biocathodes can be used to produce useful products or remove unwanted compounds from the environment [244]. The presence of conductive polymers in these systems can both improve the rate of electron transport and reduce the biofouling of electrodes, increasing the duration of their effective operation. For example, in [245], a composite of polypyrrole with poly(methylene blue) was used in a cathode half-cell with the photosynthetic microalgae *Chlorella vulgaris*. The developed biocathode demonstrated the highest short-circuit current density of 65 mA/m^2^ compared to the variants without microalgae or without the conductive polymer. Two years later, Zhang et al. [246] proposed modifying the biocathode with the mineral tourmaline, which was retained on the surface by PANI, and microorganisms from sludge served as the biocatalyst.

One of the most frequently used enzymes in biocathodes is laccase [247,248,249]. The encapsulation of laccase in conducting polymers is a fairly well-known method [250], which was subsequently applied in fuel cells. Commercially available polymers [200,251], polyvinyl alcohol [252], polypyrrole [193], etc., are used as an immobilizing agent for laccase. It should be noted that the choice of suitable polymers for laccase immobilization is hampered by the fact that laccase has multiple redox centers, and its orientation relative to the electrode has a great influence on electron transfer [253,254]. In particular, the authors of [249] compared three methods of laccase immobilization on a biocathode: crosslinking with electropolymerized PANI, entrapment in copper alginate, and encapsulation in Nafion micelles. PANI-immobilized laccase showed the highest stability and activity, producing a power density of 38 ± 1.7 mW m^−2^. Shrier et al. [255] used laccase and anthracene-modified MWCNTs in combination with various immobilization polymers to increase the stability of biocathodes. From the presented results, it can be seen that the Nafion polymer modified with tetrabutyl ammonium bromide provides a more reliable immobilization strategy compared to other tested options. Finally, in [256], laccase from *Trametes hirsuta* was immobilized into poly(3,4-ethylenedioxythiophene) and polyaniline polymer matrices for biofuel cell application. These biocathodes had a high open-circuit cathode potential and generated high current densities reaching 1 mA cm^−3^ at 0.45 V. However, these biocathodes were found to be extremely pH-sensitive, reaching maximum power only in a very narrow working pH range, with the optimal value of 4.2.

As enzymes for biocathodes, it is proposed to use not only laccase but also glucose oxidase or horseradish peroxidase. In [257], glucose oxidase was immobilized in poly(pyrrole-2-carboxylic acid) together with Prussian blue on the surface of a graphite electrode. GOx molecules were covalently attached to the carboxyl groups of PPCA by an amide bond, and the presence of Prussian blue allowed this electrode to effectively reduce H_2_O_2_ formed during GOx-catalyzed glucose oxidation. In general, it can be recognized that the use of conductive polymers as a method of immobilization for enzymes on biocathodes is not yet as widespread as compared to the classical adsorption or covalent binding of an enzyme.

As can be seen from the works we have reviewed in this section, the main attention of the investigators is aimed at the use of biopolymers in the composition of anodes and cathodes of microbial BFCs in order to increase the specific power and the generated voltage of BFCs. This goal is achieved by increasing the effective surface area of bioelectrodes when using conductive polymers, as well as by the more reliable immobilization of biocatalysts in polymer gels. It can be noted that about 80% of the works to some extent use composite materials based on conductive polymers with nanomaterials and nanoparticles as part of bioanodes and biocathodes. At the same time, the number of effectively used conductive polymers in the composition of BFCs is still quite limited: for the most part, researchers use PEDOT:PSS, PANI, and PPy. Nevertheless, each of these conductive polymers has its drawbacks, so one of the important areas of research should be the synthesis of new conductive polymers and composites based on them for reliable and long-term use in the composition of biofuel elements. Nevertheless, it is obvious that the use of conductive polymers is one of the most effective and inexpensive ways to increase the power and duration of operation of biofuel elements; therefore, the share of works devoted to this topic will only increase.

**Table 4 polymers-15-03783-t004:** Examples of bioanodes for enzymatic and microbial biofuel cells based on commonly used conductive polymers.

Polymer	Anode Composition	Biocatalyst	Substrate	Real Application	Power	Current Density	Maximum Voltage, mV	Reference
PEDOT	PEDOT/MWCNT/ GOx	Glucose oxidase (anode)/bilirubin oxidase (cathode)	Glucose	Animal/human body implantation	236 mW cm^−2^	350 mA cm^−2^	620	[192]
PEDOT	Biochar/NiFe_2_O_4_/ PEDOT/bacteria	Pre-acclimated bacteria from an MFC reactor	Glucose	Sustainable green energy generation from wastewater	1200 mW m^−2^	3324 mA m^−2^	690	[223]
PEDOT	PEDOT/graphene/ nickel	*Escherichia coli*	Glucose	Generating energy from organic wastes	0.32 mW cm^−2^	1.7 mA/cm^−2^	210	[224]
PEDOT	Carbon felt/ PEDOT/bacteria	Microbial consortium	Glucose	Sewage wastewater treatment	2.864 mW m^−2^	3813 mA m^−2^	1470	[206]
PEDOT:PSS	PEDOT:PSS/ sulfonated graphene oxide/ ferritin/GOx	Glucose oxidase	Glucose	Self-powered glucose biosensors	-	27 ± 2 mA cm^−2^	-	[188]
PEDOT:PSS	Carbon veil/ PEDOT:PSS/ sludge	Sludge	Urine	MFC continuously fed with neat human urine	10.70 µW∙cm^−2^	200 μA∙cm^−2^	705	[225]
PEDOT:PSS	PEDOT:PSS/ graphene/Nafion/*G. oxydans*	*Gluconobacter oxydans*	Synthetic/ municipal wastewater	Treatment of municipal wastewater samples with low pH	81 mW m^−2^	2.1 mA cm^−2^	550	[227]
PEDOT:PSS	Carbon felt/ PEDOT:PSS/thermally expanded graphite/bacteria	Microbial consortium	Sewage wastewater/ glucose	Sewage wastewater treatment	68.7 mW m^−2^	969.3 mA m^−2^	540	[228]
PEDOT:PSS	Nickel foam/MgCoO_2_/PEDOT:PSS/bacteria	Sludge	Wastewater	New materials for wastewater treatment systems	494 mW m^−2^	900 mA m^−2^		[229]
Rectangular polypyrrole	Nickel foam/Nafion/GOx/polyvinylpyrrolidone/polypyrrole	Glucose oxidase/laccase	Glucose	DET anode for glucose fuel cells	0.350 mW cm^−2^	3.1 mA cm^−2^	1160	[194]
Polypyrrole	Cellulose/ polypyrrole/FDH	Fructose dehydrogenase (anode)/laccase (cathode)	Fructose	Use of hybrid capacitive polymer materials in BFC	2.1 mW cm^−2^	13 mA cm^−2^	590	[195]
Polypyrrole	Graphite/PPy/yeast	* Saccharomyces cerevisiae *	Glucose	Evaluation of the possibility of using yeast in BFC	47.12 mW m^−2^	5.2 mA cm^−2^	390	[215]
Polypyrrole	Stainless steel/PPy/bacteria	Sludge from fruit wastewater treatment	Acetate	Creation of cheap BFC bioanodes	1190.94 mW m^−2^	1366.4 mA m^−2^	547	[216]
Polypyrrole	Carbon black/PPy/carboxymethyl cellulose/CNTs/bacteria	Electricity-producing microorganisms	Acetate	Environmentally friendly modification of composite anode	2970 mW m^−2^	5.20 A m^−2^	-	[218]
Polypyrrole	Carbon felt/PPy/Fe_3_O_4_/ bacteria	Electricity-producing bacteria from soil	Molasses wastewate	Degrading molasses wastewater	-	0.170 A m^−2^	-	[219]
Polypyrrole	Carbon cloth/PPy/bacteria	-	-	Capacitive bioanode for paper-based microbial fuel cell	29 µW cm^−2^	299 μA cm^−2^	580	[220]
Sigracell^®^ PV15	PV15/diethylenetriamine/glutaraldehyde/GOx	Glucose oxidase (anode)/laccase (cathode)	Glucose	Conversion of organic substrates contained in wastewater of oil mills	2.41 µW cm^−2^	2.8 μA cm^−2^	390	[200,202]
Polyaniline	Polyaniline/ferritin/GOx	Glucose oxidase	Glucose	One-step electrode construction BFC anodes	-	22.3 ± 2 mA cm^−2^	-	[196]
Polyaniline	GCE/Au@PANI/GOx	Glucose oxidase	Glucose	High-throughput membrane-less bioenergy devices	685 µW cm^−2^	12 mA cm^−2^	760	[197]
Polyaniline	Nafion/PANI1600@CNTs/GOx	Glucose oxidase (anode)/laccase (cathode)	Glucose	DET anode/cathode for glucose fuel cells	1.12 mW cm^−2^	6.2 mA cm^−2^	780	[198]
Polyaniline	Nickel foam/graphene oxide/PANI/GOx	Glucose oxidase	Glucose	Flow-through electrodes for glucose-based enzymatic microfuel cells	118 µW cm^−2^	-	-	[199]
Polyaniline	Chitosan@reduced graphene oxide/polyaniline/ferritin/GOx	Glucose oxidase	Glucose	Glucose-based EFCs	-	3.5 mA·cm^−2^	-	[201]
Polyaniline	Bacterial cellulose/polyaniline/TiO_2_/*S. xiamenensis*	*Shewanella xiamenensis*	Glucose	Low-cost compact microbial fuel cells	40.66 W m^−3^	116.72 A m^−3^	790	[208]
Polyaniline	Carbon cloth/rGO/polyanliline/bacteria	Microbial consortium	Domestic wastewater	MFC for wastewater recovery	306 mW m^2^	1050 mA m^−2^	381	[210]
Polyaniline	Graphite/PANI/bacteria	Activated sludge	Potato powder/soybean powder	Purification of biodegradable organic compounds in wastewater	256.4 mW cm^−2^	324.2 mA cm^−2^	-	[211]
Polyaniline	Carbon paper/PANI/TiO_2_/graphene/Nafion/*S. oneidensis*	*Shewanella oneidensis*	Trypton	Bifunctional catalyst to improve the performance of both the anode and cathode of MFCs	79.3 mW m^2^	135 mA m^−2^	650	[235]
Polydopamine	Activated carbon/PDA/bacteria	Microbial consortium	Acetate	Superhydrophilic surface for microbial anodes	803 mW m^−2^	4 A m^−2^	540	[212]
Polydopamine	Carbon felt/PDA/*S. xiamenensis*	*Shewanella xiamenensis*	Lactate	Possibility of immobilization of redox-active PDA on the surface of individual cells	452.8 mW m^−2^	142.7 μA cm^−2^	750	[214]

**Table 5 polymers-15-03783-t005:** Examples of biocathodes for enzymatic and microbial biofuel cells based on commonly used conductive polymers.

Polymer	Cathode Composition	Biocatalyst	Substrate	Real Application	Power	Current Density	Maximum Voltage, mV	Reference
PVA	Carbon cloth/PVA/WRF	White rot fungi	Copper-containing solution	Removing Cu^2+^ from the wastewater	41.3 mW m^−2^	260 mA m^−2^	710 mV	[171]
PVA	CNTs/PVA/sorbitol/laccase	Laccase	ABTS/O_2_	Printable laccase-based biocathode for fuel cell applications	11 µW∙cm^−2^	50 mA cm^−2^	1100	[252]
Polyindole	Polyindole/iron phthalocyanine/CNTs	-	Domestic wastewater/acetate	Low-cost composite with high ORR	799 mW m^−2^	3480 mA m^−2^	695	[236]
Polypyrrole	Carbon cloth/PPy	*Shewanella putrefaciens* (in anode)	Glucose	Iron removal from wastewater	190 mW m^−2^	1.27 A m^−2^	690	[237]
Polypyrrole	Carbon paper/Ni–NiO/PPy–rGO	Mixed bacterial culture from a previously used MFC	-	COD removal	678.79 mW m^−2^	2134.56 mA m^−2^	610	[238]
Polypyrrole	Stainless steel/PPy/poly(methylene blue)/C. vulgaris	*Chlorella vulgaris* (cathode)/g *Saccharomyces cerevisiae* (anode)	Carbon dioxide	A photosynthetic biocathodic half-cell	7 mW m^2^	65 mA m^−2^	370	[239]
Poly(pyrrole-2-carboxylic acid)	Graphite rod/PPCA/Prussian blue/GOx	Glucose oxidase	Glucose	Biocathode for glucose-powered single-enzyme biofuel cell	-	31.68 μA cm^−2^	430	[257]
Polyaniline	Carbon felt/PANI/sludge	Aerobic sludge	NaHCO_3_	Utilization of nitrates	199 mW m^2^	1420 mA m^−2^	482	[181]
Polyaniline	Graphite/PANI/tourmaline/bacteria	Sludge	NaHCO3	Mineral-based biocathode	266 mW m^−2^	1220 mA m^−2^	-	[246]
Polyaniline	PANI/laccase	Laccase	ABTS/O_2_	Biocathode for azo dye decolorization	38 mW m^−2^	175 mA m^−2^	-	[249]

## 6. Conclusions

Currently, the main attention in the creation of bioelectrochemical devices is given to the stage of immobilization of the biocatalyst on the surface of the signal transducer. The materials used at this stage must firmly hold the biocatalyst in contact with the working electrode and provide free access to the analyte without letting aggressive reagents through. The most important characteristics of immobilizing agents are biocompatibility, high electrical conductivity, and chemical stability. Conductive polymers, as shown above, meet all of the above requirements and therefore are increasingly used in bioelectronics. In addition, the advantages of conducting polymers include their ability to change properties depending on the method and conditions of preparation. The introduction of various modifiers into the structure of conducting polymers also makes it possible to change and vary the properties of the obtained composites. For example, polymers with relatively low conductivity can be modified with carbon nanomaterials or metal nanoparticles [258].

Most of the biocompatibility problems of conductive polymers can be overcome by combining them with Ag nanoparticles or other biodegradable polymers [4]. An important criterion for the use of conductive polymers is the economic component—the ease of synthesis, the availability of precursors, and the absence of the need for expensive equipment. An interesting area of research is the initialization of the polymer synthesis process using enzymes or microorganisms while simultaneously capturing them with a growing polymer film on the electrode surface. Such one-stage formation of the biosensor makes it possible to simplify the measurement scheme and reduce the analysis time. The current need is focused mainly on biosensors that can be used in real samples to quickly diagnose pathogens. The most significant advances in this field can be expected, and composites with metal nanoparticles will probably play the leading role due to the increase in the sensitivity of nanocomposites due to the stable p-n heterojunction formed between the p-type polymer and n-type metal oxide nanoparticles. Metal nanoparticles are also suitable materials for stable sensors with good repeatability and reproducibility for the simultaneous detection of biomolecules or drugs in real samples due to their electrocatalytic properties. Along with traditional one-polymer composites, researchers are currently creating materials from two or more polymers, one of which may be non-conductive but used as a stabilizer for the structure of the final composite. Using this approach, new classes of highly sensitive multifunctional materials can be created that will be used to immobilize biological objects as part of various bioelectrochemical devices.

## Figures and Tables

**Figure 1 polymers-15-03783-f001:**
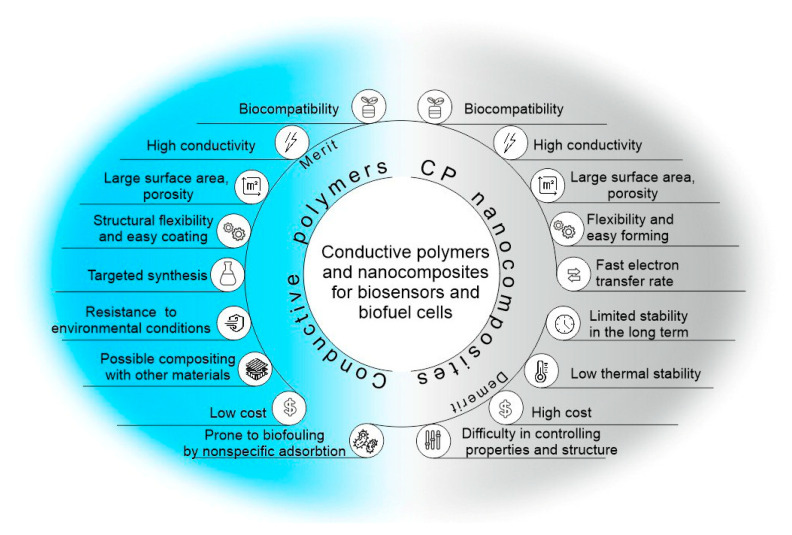
Advantages and disadvantages of conducting polymers for use in biosensors and biofuel cells.

**Figure 2 polymers-15-03783-f002:**
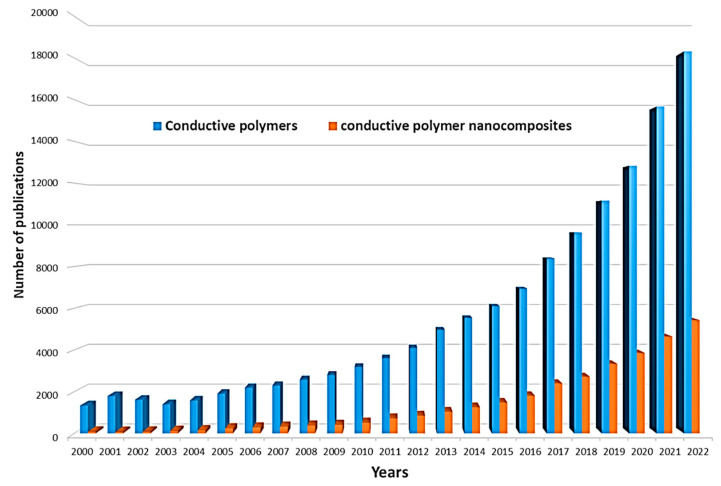
Number of publications in the ScienceDirect database per year with conductive polymers and conductive polymer nanocomposite as keywords.

**Figure 3 polymers-15-03783-f003:**
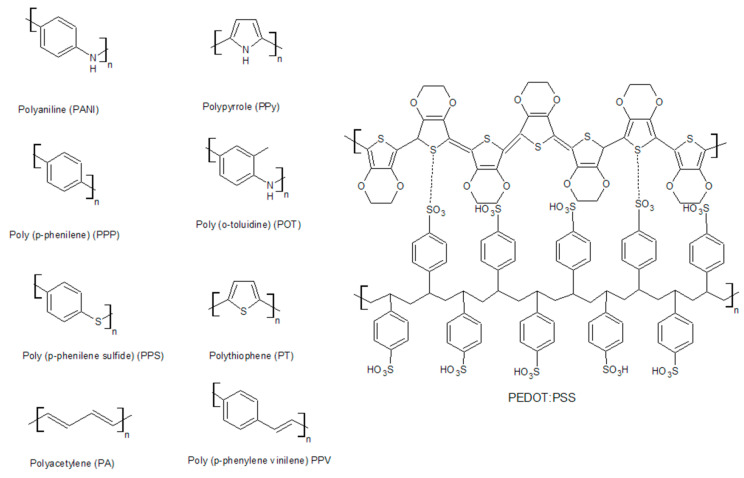
Structural forms of the most common conductive polymers.

**Figure 4 polymers-15-03783-f004:**
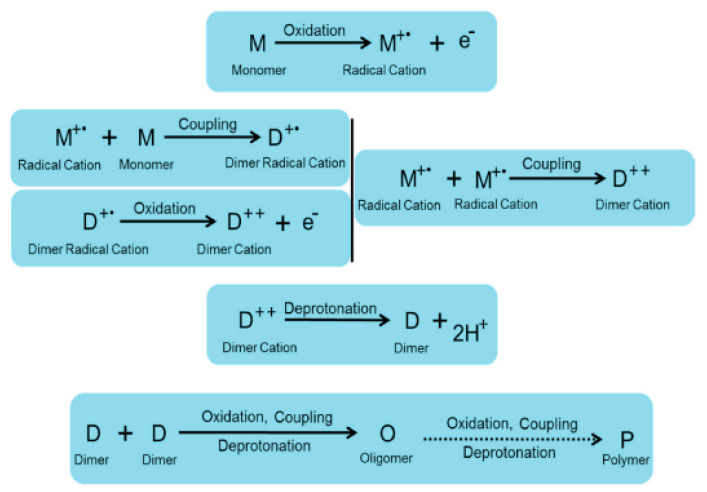
General mechanism for the formation of conductive polymers. Reproduced with permission from ref. [28]. Copyright 2021 Elsevier Ltd.

**Figure 5 polymers-15-03783-f005:**
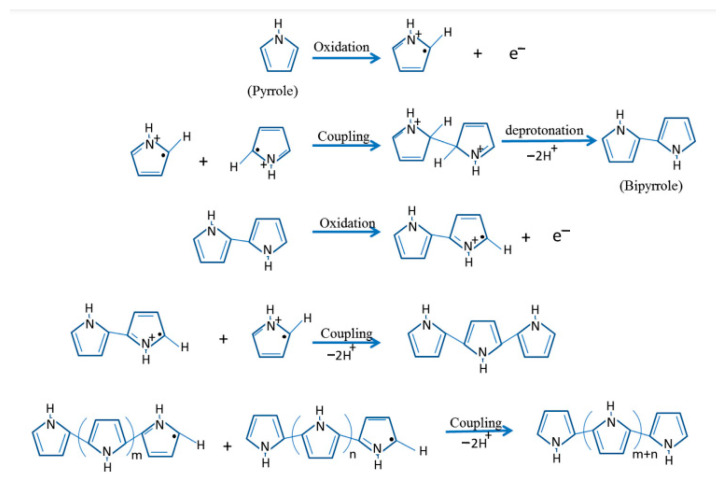
Mechanism of chemical oxidative polymerization of polypyrrole. Reproduced with permission from ref. [40]. Copyright 2020 John Wiley & Sons Ltd.

**Figure 6 polymers-15-03783-f006:**
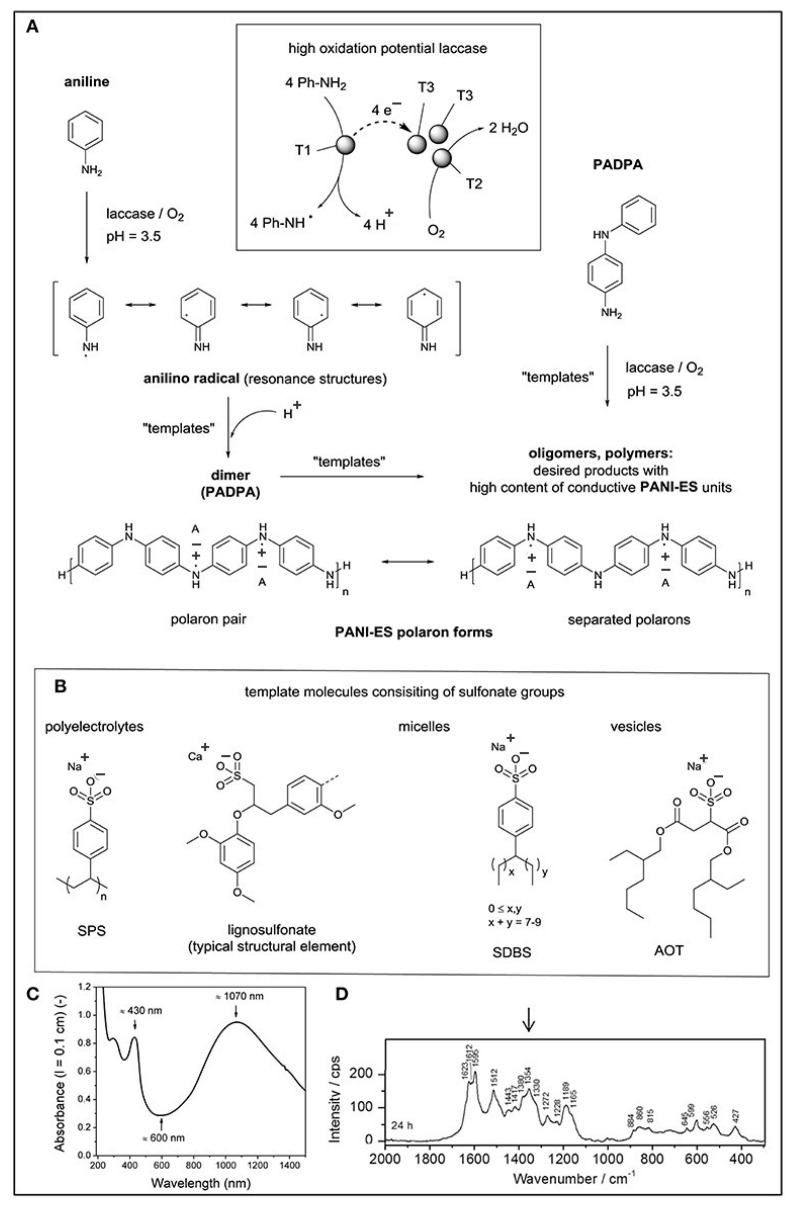
(**A**–**D**) Synthesis of polyaniline using laccase. Reproduced with permission from ref. [42]. Copyright 2019 The Authors, Frontiers.

**Figure 7 polymers-15-03783-f007:**
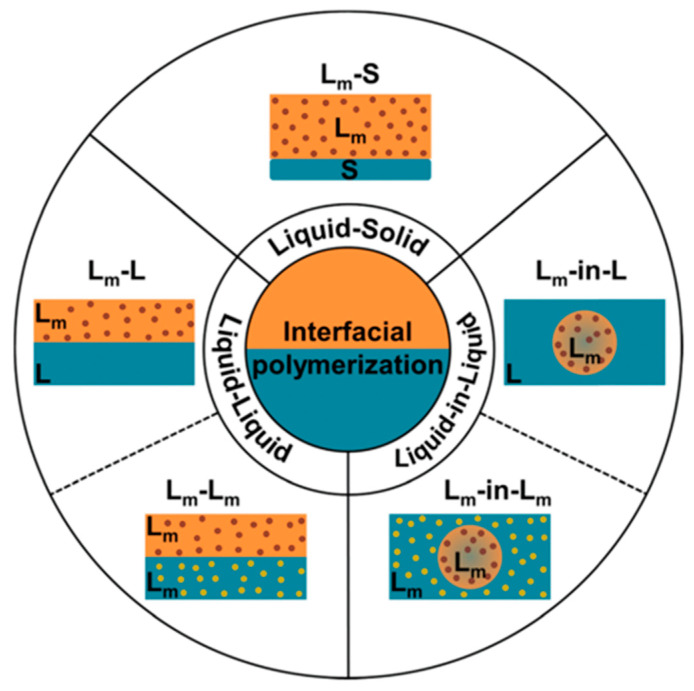
Typical schemes of interfacial polymerization (Lm—liquid phase containing monomers; S—solid phase; L—liquid phase) Reproduced with permission from ref. [43]. Copyright 2017 The Authors, Royal Society of Chemistry.

**Figure 8 polymers-15-03783-f008:**
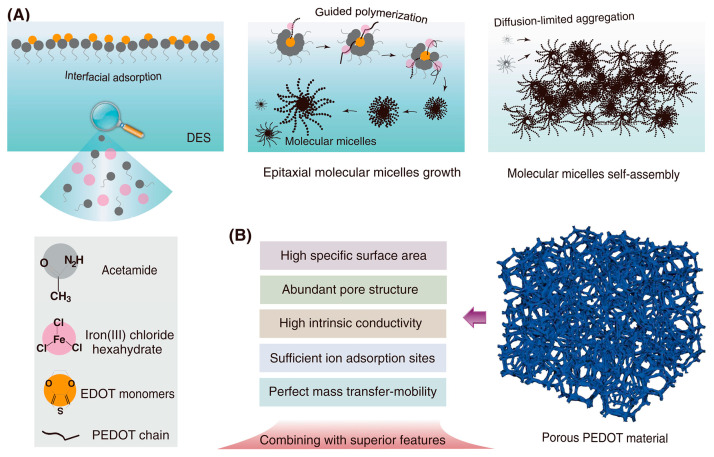
Scheme for the synthesis of porous poly(3,4-ethylenedioxythiophene) (PEDOT). (**A**) Design of porous PEDOT based on interfacial polymerization in a deep eutectic solvent system. (**B**) Some unique features of PEDOT porous material. Reproduced with permission from ref. [44]. Copyright 2023 The Authors, Wiley.

**Figure 9 polymers-15-03783-f009:**
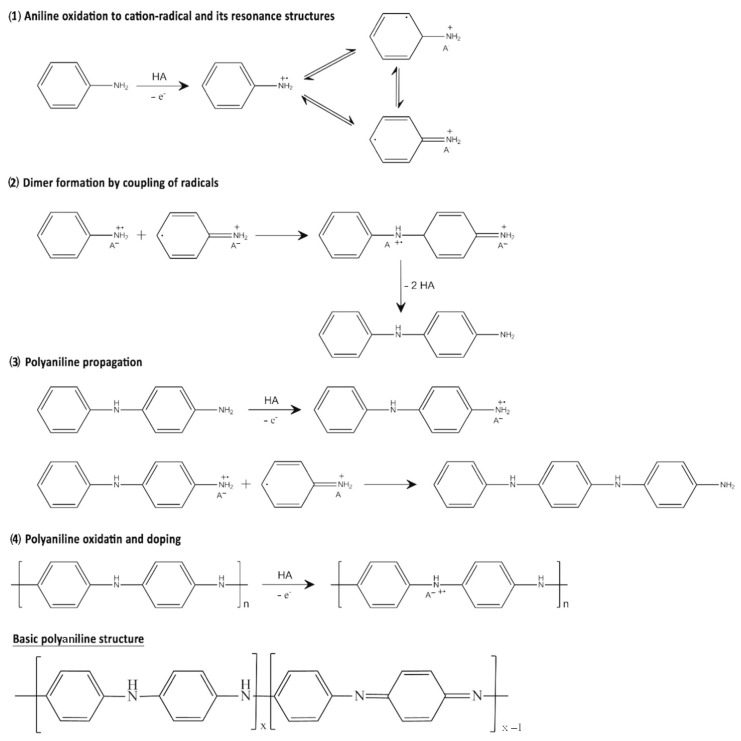
Mechanism of aniline electropolymerization. Reproduced with permission from ref. [50]. Copyright 2020 The Authors, IOP.

**Figure 10 polymers-15-03783-f010:**
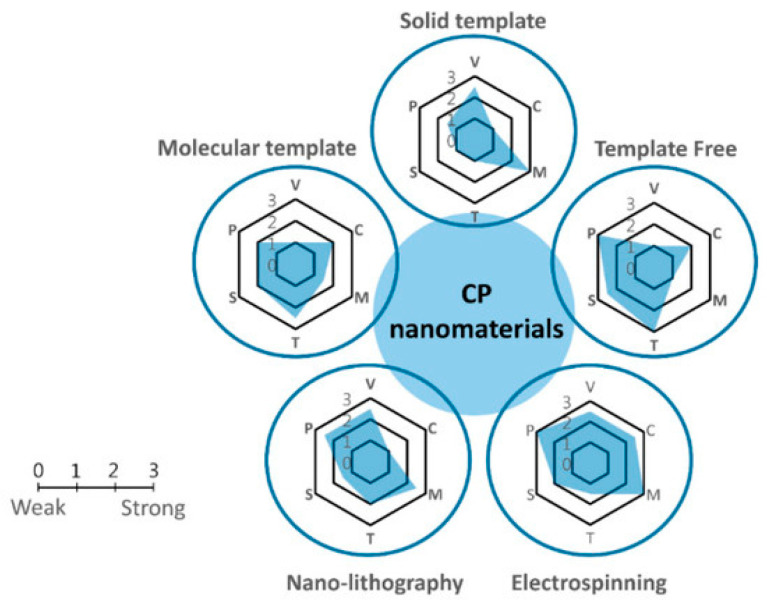
Methods for the synthesis of nanosized conductive polymers and nanocomposites. Reproduced with permission from ref. [34]. Copyright 2016 The Authors, MDPI.

**Figure 11 polymers-15-03783-f011:**
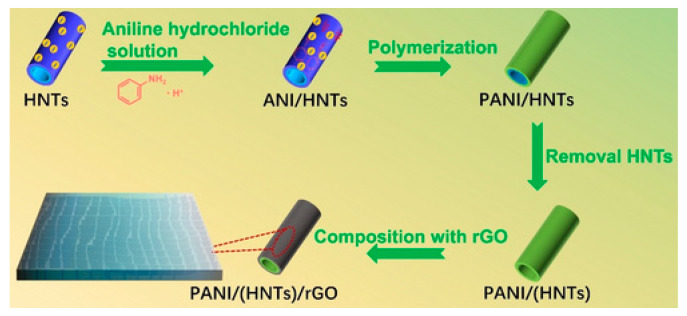
Preparation scheme for PANI(HNTs) and PANI(HNTs)/rGO. Reproduced with permission from ref. [60]. Copyright 2022 John Wiley and Sons.

**Figure 12 polymers-15-03783-f012:**
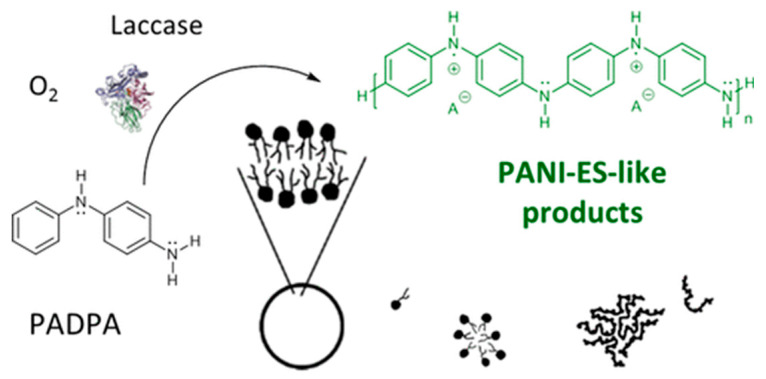
Scheme for the synthesis of polyaniline using soft templates. Reproduced with permission from ref. [61]. Copyright 2019 The Authors, American Chemical Society.

**Figure 13 polymers-15-03783-f013:**
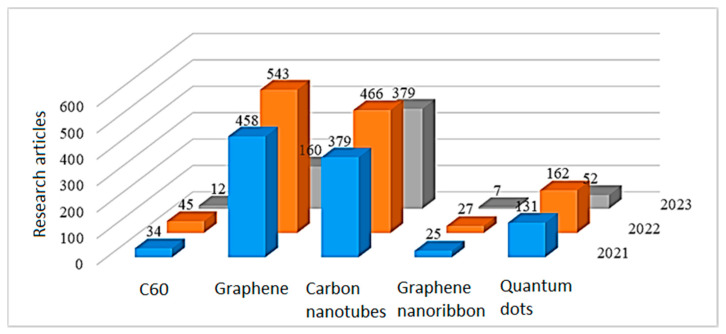
Number of experimental publications related to the development of biosensors based on “conductive polymers and carbon nanomaterials” composites. ScienceDirect database and keywords “conductive polymer-biosensor”.

**Figure 14 polymers-15-03783-f014:**
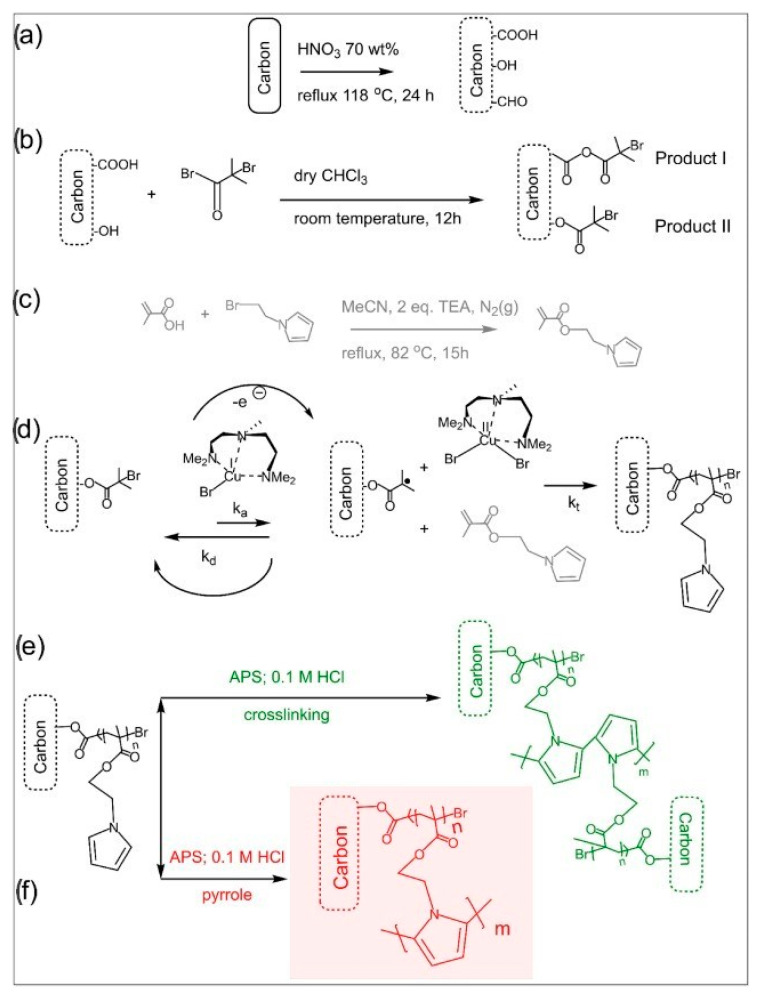
Method of radical polymerization with atom transfer to create a nanostructured material. (**a**) Modification of multi-walled carbon nanotubes; (**b**) addition of α-bromoisobutyryl bromide; (**c**) synthesis of N-ethylpyrrole methacrylate monomer; (**d**) modification of the monomer with brominated carbon nanobooks; (**e**) polymerization without the addition of pyrrole; (**f**) copolymerization of the modified polymer with pyrrole. Reproduced with permission from ref. [66]. Copyright 2022 The Authors, Elsevier.

**Figure 15 polymers-15-03783-f015:**
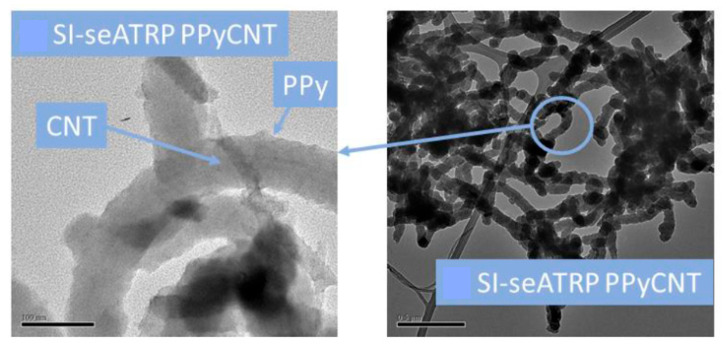
Nanostructured material based on carbon nanotubes and polypyrrole [66]. Copyright 2022 The Authors, Elsevier.

**Figure 16 polymers-15-03783-f016:**
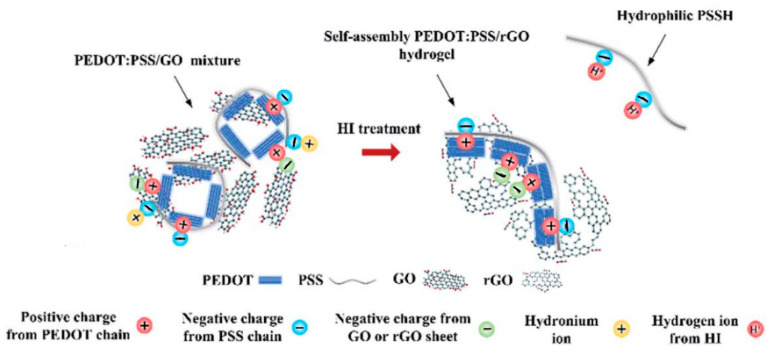
Three-dimensional porous composite based on poly(3,4-ethylenedioxythiophene) (PEDOT), poly(4-styrenesulfonate) (PSS), and graphene. Reproduced with permission from ref. [67]. Copyright 2020 Royal Society of Chemistry.

**Figure 17 polymers-15-03783-f017:**
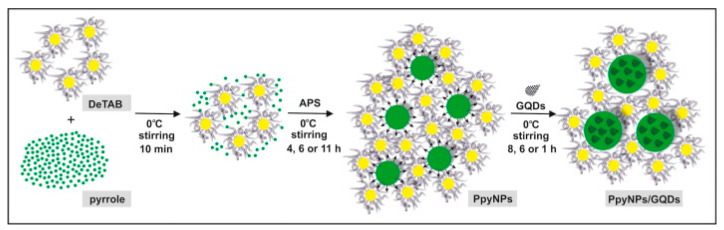
Composite based on carbon nanodots and polypyrrole. Reproduced with permission from ref. [68]. Copyright 2022 The Authors, Elsevier.

**Figure 18 polymers-15-03783-f018:**
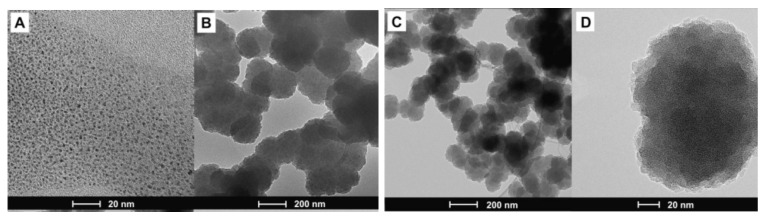
Microscopy of the starting materials and the resulting polymer: (**A**)—carbon nanodots; (**B**,**C**)—polypyrrole; (**D**)—hybrid. Reproduced with permission from ref. [68]. Copyright 2022 The Authors, Elsevier.

**Figure 19 polymers-15-03783-f019:**
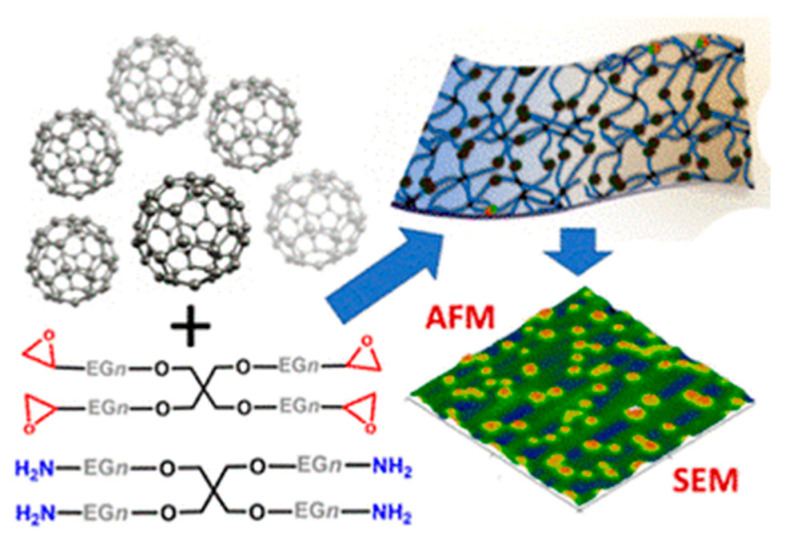
Fullerene-based ultrathin films. Reproduced with permission from ref. [71]. Copyright 2023 The Authors, American Chemical Society.

**Figure 20 polymers-15-03783-f020:**
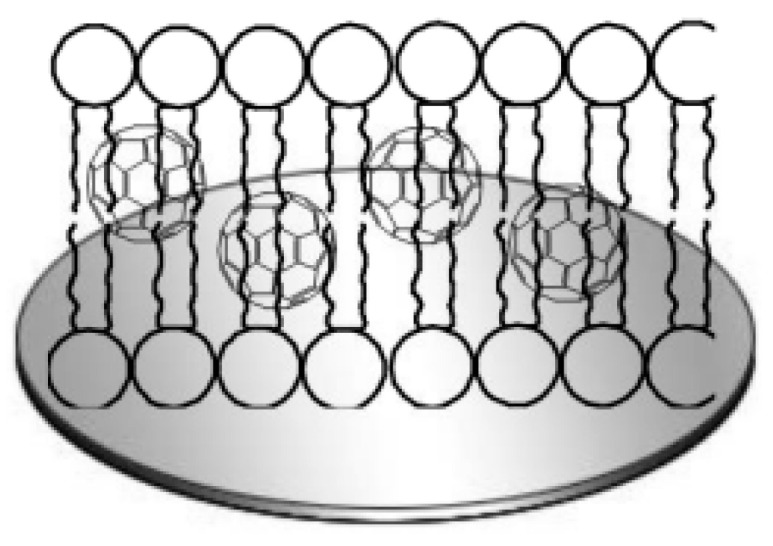
Bilayer membrane for lactamase immobilization. Reproduced with permission from ref. [73]. Copyright 2020 Elsevier.

**Figure 21 polymers-15-03783-f021:**
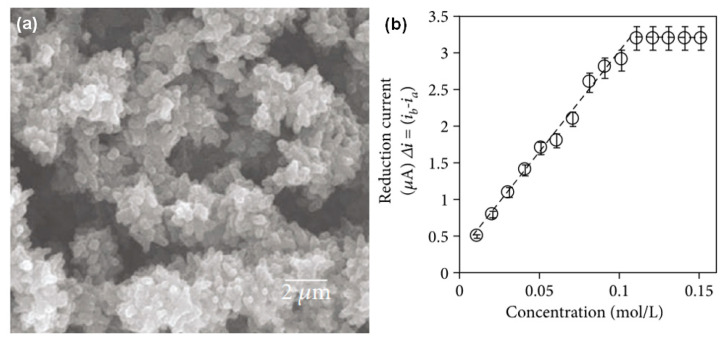
(**a**)—SEM image of PANI; (**b**)—response of biosensor: reduction current vs. concentration of glucose. Reproduced with permission from ref. [90]. Copyright 2021 The Authors, Hindawi.

**Figure 22 polymers-15-03783-f022:**
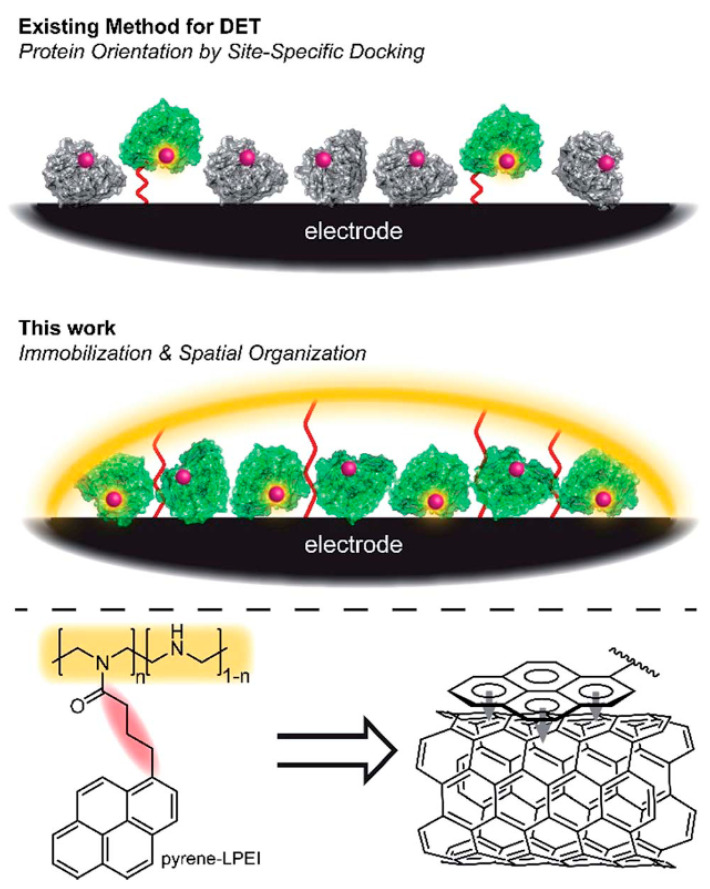
Scheme depicting the docking approach as compared to the hydrogel immobilization approach described here. Active redox protein is depicted in green, while inactive/denatured redox protein is depicted in gray. Reproduced with permission from ref. [100]. Copyright 2018 Royal Society of Chemistry.

**Figure 23 polymers-15-03783-f023:**
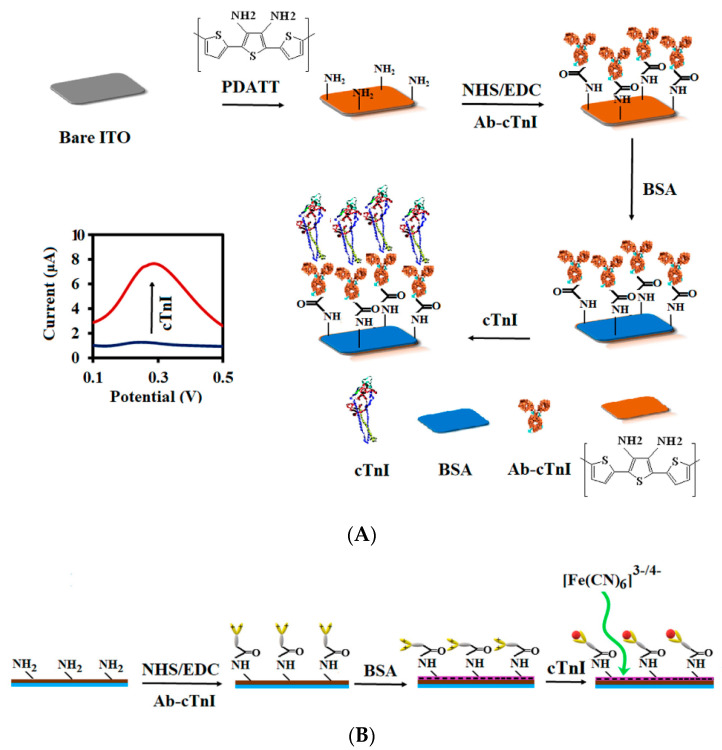
(**A**) Schematic construction step of ITO/PDATT/cTnI-Ab/BSA electrochemical immunosensor for detection of cTnI. (**B**) Change in the orientation of the cTnI antibody molecules after their attachment to PDATT, backfilling with negatively charged BSA and binding to the cTnI protein. The green arrow depicts the flow of the [Fe(CN)_6_]^3−/4−^ ions through the gaps between the upward-oriented antibody molecules. Reproduced with permission from ref. [132]. Copyright 2021 Elsevier.

**Table 1 polymers-15-03783-t001:** Electrical properties of some widely used conductive polymers [28,29,30].

Conductive Polymer	First Synthesized	Molecular Weight of the Monomer (g mol^−1^)	Conductivity Type	Specific Capacitance (Fg^−1^)	Conductivity (S cm^−1^)
Polyacetylene	1977	26	n, p	241	10^3^–1.7 × 10^5^
Polypyrrole	1979	67	P	530	10^2^–7.5 × 10^3^
Polyparaphenylene	1979	78	n, p	-	10^2^–10^3^
Polyparavinylene	1979	28	P	-	3–5 × 10^3^
Poly(3,4-ethylenedioxythiophene)	1980	142	n, p	92	300
Polyaniline	1980	93	n, p	240	30–200
Polythiophene	1981	84	p	485	10–10^3^

## Data Availability

Data sharing not applicable.

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
