# Peer review of "Conductive Polymers and Their Nanocomposites: Application Features in Biosensors and Biofuel Cells"

_polymers, 2023, doi:10.3390/polym15183783_

Round 1
Reviewer 1 Report
In this manuscript, the authors reviewed conductive polymers in the literature and their applications in biosensors and bio-fuel cells. The authors started from the synthesis of the polymers and the related polymer/nanomaterial composite. The synthetic mechanisms and polymerization preparations of conductive polymers were well summarized. After introducing the nanomaterials, these conductive composites can also be used for biosensing or other electrochemical applications. The authors well described the electrochemical mechanisms of these applications and well summarized the comparison in tabular forms. Overall, the review is well written with good organization. It is thus recommended for publication. Some minor issues are listed below for authors to perfect the manuscript as follows:
1. Several abbreviations in the tables should be addressed as footnotes. For example, in table 1, the column for conductivity type, what does it mean (n,p)? In table 2, DPV, EIS, CV…
2. Table 2 is too massive. It would be better to have a separate table for each sub-section, such as enzyme, microbial sensors, etc. It would be easier for reader to follow both text and the related examples.
3. Table 4 is too massive. Can the authors divide it in to sections according to the anodes or cathodes like those in section 5.1 and 5.2? It would be better to have a separate table for each sub-section, such as enzymatic anodes, etc.
NA
Reviewer 2 Report
The presented manuscript is rich in content. The authors spent a lot of time to ensure that it was carefully prepared and formed a logical whole. The review is based on 258 publications. Paragraphs are well-described and contain the most important information on a particular aspect.
Minor language correction is required.
Reviewer 3 Report
This review paper provides a comprehensive overview of an important topic. The authors effectively highlight the significance of conductive polymers in biosensors and biofuel cells, showcasing their potential applications. The paper's structure and content are well-organized and informative. However, minor revision is required to perform before its publication:
· Check this manuscript to avoid English language errors carefully, especially grammar and spelling errors, typos, extra space, superscripts and subscripts, etc. e.g., line 603, 687, line 467, and so on
· Please carefully review the units in Table 4, there are so many errors. And since the column is called “Current density”, then there shouldn’t be any value as ampere.
Reviewer 4 Report
Dear Authors,
This is a well prepared review about conductive polymers and their nanocomposites. However before publication I would suggest some minor revisions:
A more detailed future perspectives should be included into your manuscript. It may be with a separate section or within an appropriate one. Together with this an important part of this future percpective should be included within the abstract.
Quality of the figures should be improved and fonts of all figures should be uniform. Some are not readable like in fig14.
Regards
